# Beach Profile, Water Level, and Wave Runup Measurements Using a Standalone Line-Scanning, Low-Cost (LLC) LiDAR System

Christopher S. O'Connor 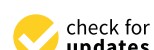 and Ryan S. Mieras *

Department of Physics and Physical Oceanography, University of North Carolina Wilmington, Wilmington, NC 28403, USA
* Correspondence: mierasr@uncw.edu

**Abstract:** A prototype rapidly deployable, Line-scanning, Low-Cost (LLC) LiDAR system (USD 400 per unit; 2020) was developed to measure coastal hydro-morphodynamic processes. A pilot field study was conducted at the U.S. Army Corps of Engineers, Field Research Facility (FRF) in Duck, North Carolina, USA to evaluate the efficacy of the LLC LiDAR in measuring beach morphology, wave runup, and free-surface elevations against proven approaches. A prototype LLC LiDAR collected continuous cross-shore line scans for 25 min of every half hour, at ~7 revolutions/s and ~1.3° angular resolution, at two locations (one day at each location), spanning 12 m (i) on the backshore berm (35 scans; Series B) and (ii) in the swash/inner surf zone (28 scans; Series C). LLC LiDAR time-averaged beach profiles and wave runup estimates were compared with the same quantities derived from the continuously sampling terrestrial LiDAR scanner installed atop the dune at the FRF (DUNE LiDAR). The average root-mean-square difference (RMSD) between 17 (6) time-averaged LLC and DUNE LiDAR beach profiles was 0.045 m (0.031 m) with a standard deviation of 0.004 m (0.002 m) during Series B (Series C). Small-scale (cm) swash zone bed level changes were resolved over 5-min increments with the LLC LiDAR. The RMSD between LLC- and DUNE LiDAR-derived wave runup excursions over two 25-min segments was 0.542 m (cross-shore) and 0.039 m (elevation) during the rising tide and 0.366 m (cross-shore) and 0.032 m (elevation) during the falling tide. Between 72–79% of the LLC LiDAR wave runup data were more accurate than the RMSD values, thereby demonstrating the LLC LiDAR is an effective, low-cost instrument for measuring wave runup and morphodynamic processes. Co-located water levels were measured with a continuously sampling (16 Hz) RBR*solo*$^3$ D|wave16 pressure logger during Series C. LLC LiDAR free-surface elevations at the nadir during one high tide (4.5 h) compared well with pressure-derived free-surface elevations (RMSD = 0.024 m, $R^2$ = 0.85).

**Keywords:** embedded systems; terrestrial LiDAR; rapid response; beach morphology

## 1. Introduction

The destructiveness of tropical cyclones has been increasing over the past several decades, which may be attributed to longer lived, more energetic storms [1], and to a higher frequency of occurrence and magnitude of rapid intensification [2,3]. This increase in destructiveness, coupled with rising coastal populations [4] and infrastructure demands [5], contribute to large economic cost of these storms [6]. Hurricanes Harvey, Irma, and Maria caused more than USD 200 billion in the 2017 hurricane season alone [7,8].

Coastal scientists, engineers, and municipal planners require numerical models to predict how coastal communities may be impacted by storm surges and waves, and the resulting sediment transport, to improve the resilience of these communities and to reduce the costs of coastal storms [7,9,10]. Many of these numerical models rely on in situ measurements before, during, and after storm impact for validation [11–13]. Extreme

hydrodynamic conditions make collecting in situ measurements of beach profile evolution and total water levels during major storms both difficult and dangerous [14,15].

Advancing our understanding of storm processes and impacts will require novel instrument platforms for better observations of sediment transport processes and hydro-morphodynamics [7]. Traditional methods for surveying beach profile elevation and shoreline evolution have a low temporal resolution, with days, weeks, or months between measurements, due to logistical and economic constraints [15–17], and can typically only be employed before and after a storm, but not during, when the most rapid morphological changes occur [18]. Developing a more complete understanding of storm processes requires sensors capable of (a) continuously measuring beach profile evolution and total water levels (wave runup, wave setup, tide, and storm surge) before, during, and after a storm with higher spatial and temporal resolution than traditional methods, and (b) withstanding high wind, energetic waves, and/or sediment accretion/erosion [19]. Fully standalone, self-contained systems with low infrastructure requirements are necessary for rapid-deployability when predictions of location(s) that may experience the most significant impact(s) are uncertain or widespread. Low cost is also a high priority, due to potential loss of equipment during storm impact [3,20].

Terrestrially based LiDAR systems have proven useful for measuring wave energy [21], shape [22], transformation [23], setup [24], and runup [15] across the surf and swash zones, as well as beach morphology [15,25–27]. Alternatively, ultrasonic sensors have also been used to measure beach elevation changes over time, before, during, and after storm impact [19,28]. Modern terrestrially based LiDAR systems are still largely cost prohibitive and require a level of infrastructure that makes rapid deployment unfeasible or impractical, particularly on remote barrier islands which may serve to protect the hinterland [18]. A prototype rapidly deployable Line-scanning, Low-Cost (LLC) LiDAR system was developed in 2020 at the University of North Carolina Wilmington to provide an affordable method of collecting beach profile and free-surface elevation data with reasonably high spatial and temporal resolution and minimal infrastructure requirements.

The design and operation of the prototype LLC LiDAR system is described in Section 2. A pilot field deployment where the LLC LiDAR was tested against proven terrestrial LiDAR and pressure sensor technology is discussed in Section 3. Results from the pilot field deployment are summarized in Section 4, focusing on the LLC LiDAR capability in measuring beach profiles and morphology (4.1), wave runup (4.2), and free-surface elevations (4.3).

## 2. Materials and Methods

### 2.1. Design and Operation

The prototype LLC LiDAR consists of six major components: [1] Slamtec RPLiDAR A1M8 (R5) 360° laser range scanner (USD 110); [2] Raspberry Pi 3B+ single board computer and 32 GB microSD card (~USD 40, in 2020); [3] Anker PowerCore+ 26,800 mAh PD USB battery pack (USD 140); [4] 3D printed (PLA), custom-designed mounting bracket (<USD 2); [5] Polycase enclosure (Part # WC-41) (USD 45); [6] Polycase aluminum baseplate (Part # WX-42) (USD 13). The hole pattern aligned with the 3D printed mounting bracket was manually drilled into the aluminum baseplate. Hex standoffs were used to secure the baseplate and mounting plate to each other, as well as the RPLiDAR and Raspberry Pi 3B+ to the mounting plate.

Additional components include hex standoffs and fastener hardware, 2 USB Type A to Micro B cables (M-M), 1 short HDMI cable, a rocker switch and wires, and a panel mount HDMI port. The estimated cost of the additional components is USD 50. The total cost to construct one prototype LLC LiDAR was approximately USD 400 (2020). An additional USD 50 to USD 100 is required for materials to mount or install the scanner to a pole or piling in the field (see Section 3.2).

The goal of the prototype LLC LiDAR design was to develop a scanning system that was (i) fully self-contained (i.e., battery power, data storage, and system hardware architec-

ture in one enclosure), (ii) built with as many off-the-shelf, inexpensive parts with open-source support as possible, and (iii) capable of resolving natural hydro-morphodynamic processes typical of the swash and inner surf zones [29]. The prefix *prototype* will be dropped hereafter, and only LLC LiDAR will be used when referring to the prototype system. The system was designed around the Raspberry Pi single board computer because it is capable of being powered by 5 V USB battery packs, which are widely available in numerous models, including pass-through charging, solar re-chargeable battery packs. In addition, the Raspberry Pi operating system is Linux-based (open-source) and easy to work with. The Raspberry Pi 3B+ has an on-board WiFi chip, Bluetooth, HDMI port, ethernet port, and four USB ports. Multiple USB ports allowed for simultaneous transmission of power and commands to, and point cloud data from, the RPLiDAR scanner, as well as data writing data to a USB flash drive. Remote communications via SSH and/or remote desktop (e.g., VNC) are added benefits to the Raspberry Pi computer. Lastly, future additions of components like a Real Time Clock (RTC) and Inertial Motion Unit (IMU) are possible via the General Purpose Input/Output (GPIO) pins, as well as the included $I^2C$ and SPI interfaces.

The RPLiDAR 360° laser range scanner is driven by a small 5 V motor, connected to the scanning puck with a rubber O-ring belt. The puck rotates clockwise when viewed from the same perspective as shown in Figure 1. The RPLiDAR measures distance to an object by emitting a modulated infrared laser pulse from the laser emitter and measuring the return time to the vision acquisition system. The emitter and vision acquisition system are neither axially co-aligned with each other, nor aligned axially through the puck origin. The angular resolution is not a fixed parameter with the RPLiDAR. The angular resolution depends on numerous factors including, but not limited to, rotation speed, the specifications of the data logging system (CPU, RAM, and maximum data write speeds), and most importantly, the language and efficiency of the code for logging the data. Average angular resolution, $d\theta$, was ~1.3° during testing with the Python-based data acquisition approach described later in this section. Later testing revealed that C-based data acquisition yielded slightly better angular resolution (i.e., smaller $d\theta$). The field study described in Section 3.3 used the Python-based acquisition method, so only the angular resolution from that approach is mentioned here. Despite non-fixed, non-constant $d\theta$ during data acquisition, the RPLiDAR consistently yielded near-uniform distributions of angular resolution during testing, suggesting that only minor variations from a fixed $d\theta$ occurred. In general, the standard deviation of angular resolution was relatively low (less than 0.029°). This small variation is acceptable for the application described in this manuscript.

The RPLiDAR laser wavelength is 785 nm, uses ~3 mW of power, and has a pulse length of ~110 μs, but can reach up to 300 μs. The stated minimum (maximum) scan range is 0.15 m (12 m). The actual maximum range was determined to be highly dependent upon the color (e.g., white versus black) and surface roughness (e.g., cement versus sand) of the target being scanned, as well as the ambient lighting (e.g., indoor versus outdoor and day versus night). The actual reliable range was found to be 6 to 8 m in daylight at a typical natural beach with dry beach sand. In theory, the rotation rate of the RPLiDAR scanner is configurable; but, the Python code used to operate the scanner for this LLC LiDAR model did not allow for control of the rotation rate. Tests yielded rotation rates between 6.7 to 7.1 rotations per second.

Four values are logged to a text file during a scan collection (at each rotation angle): newscan flag, quality score, range, and angle. One hour of raw, continuous point cloud scan data stored in ASCII format uses around 200 MB of storage space. The newscan flag is either a 0 or 1 (boolean) and is used to determine when a full rotation of the scanner has completed, and a new rotation is beginning. The quality score is an integer that ranges from 0 (poor) to 15 (great) and is determined by the RPLiDAR scanner for each point. The range is reported as radial distance from the center of the puck, in integer millimeters. The angle is reported as the polar angle in decimal degrees. If the scanner is oriented outdoors in a vertical position, such that a portion of the field of view is aimed towards the sky (see

Figure 1), the scanner receiver (i.e., the mirror) will not detect a return. Therefore, a quality score of 0 and a range of 0 m will be assigned.

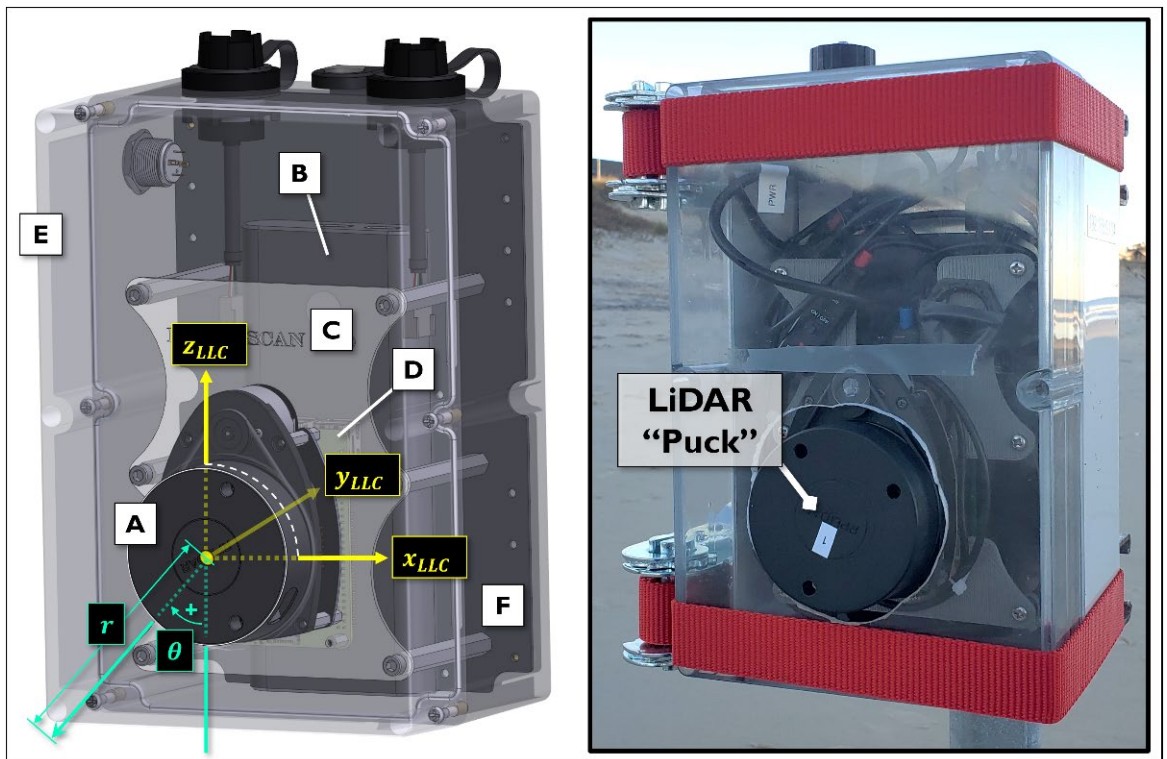

**Figure 1.** (left) 3D Computer Aided Design (CAD) model and (right) physical model, strapped to a pole on the beach, of the Line-scanning, Low-Cost (LLC) LiDAR system. The local "LLC" cartesian and polar coordinate systems are drawn on the CAD model. Major components include (**A**) Slamtec RPLiDAR A1M8 (R5) 360° laser range scanner; (**B**) Anker PowerCore+ 26,800 mAh PD USB battery pack; (**C**) 3D printed (PLA), custom-designed mounting bracket; (**D**) Raspberry Pi 3B+ single board computer; (**E**) Polycase enclosure (Part # WC-41); (**F**) Polycase aluminum baseplate (Part # WX-42).

An executable shell script calls a Python script that establishes connection with the RPLiDAR and begins retrieving data and writing to file for a user-defined duration. The duration of data collection is a command-line input read by the executable shell script. Data collection can also be automated to run on any schedule using the cron daemon (*crond*) built-in to the Raspberry Pi operating system. The start and end date and time of collection are written to the first and last rows of the ASCII data file. The dates were retrieved from the system time. The start and end times were used to linearly distribute a time stamp to each point in the point cloud, since the scanner does not have an on-board clock, and writing the actual log time of each point to file severely slowed down data collection by filling the buffer, resulting in packet loss and lower angular resolution.

### 2.2. Coordinate Transformations

The raw point cloud is measured in polar coordinates, $(r, \theta)$, and can be converted to local LLC cartesian coordinates $(x, y, z)_{LLC}$ using

$$\begin{bmatrix} x_{LLC} \\ y_{LLC} \\ z_{LLC} \end{bmatrix} = -r \cdot \begin{bmatrix} \cos\left(\frac{\pi}{2} - \theta\right) \\ 0 \\ \sin\left(\frac{\pi}{2} - \theta\right) \end{bmatrix}, \tag{1}$$

where $r$ is the radial distance to the object or surface measured by the RPLiDAR at the polar angle, $\theta$ (Figure 1). The origin, $(0, 0, 0)_{LLC}$, is located in the center of the RPLiDAR puck

(Figure 1). Positive global convention is defined as east, north, and up (normal to Earth, following the right-hand-rule convention).

Rotations about the origin of an Earth-fixed, global cartesian coordinate system are defined using Tait-Bryan angles,

$$R_{X_g} = \begin{bmatrix} 1 & 0 & 0 \\ 0 & \cos(\gamma) & -\sin(\gamma) \\ 0 & \sin(\gamma) & \cos(\gamma) \end{bmatrix}, \tag{2}$$

$$R_{Y_g} = \begin{bmatrix} \cos(\beta) & 0 & \sin(\beta) \\ 0 & 1 & 0 \\ -\sin(\beta) & 0 & \cos(\beta) \end{bmatrix}, \tag{3}$$

and

$$R_{Z_g} = \begin{bmatrix} \cos(\alpha) & -\sin(\alpha) & 0 \\ \sin(\alpha) & \cos(\alpha) & 0 \\ 0 & 0 & 1 \end{bmatrix}, \tag{4}$$

where $\gamma$ is roll, $\beta$ is pitch, and $\alpha$ is yaw. Yaw, pitch, and roll are defined as rotations about the Earth-fixed, global up ($Z_g$), north ($Y_g$), and east ($X_g$) axes. A positive rotation is defined as clockwise while "looking" in the direction of the axis of rotation (e.g., if "looking" along the east axis, $X_g$, a clockwise rotation of 45 degrees would be $\gamma = +45°$). The rotation angles ($\alpha$, $\beta$, $\gamma$) are defined from a specific LLC LiDAR initial orientation, such that all three local axes, $(x, y, z)_{LLC}$, are aligned with their global counterparts, $(x, y, z)_{ENU}$, and the local origin, $(0, 0, 0)_{LLC}$, is located at the global origin, $(0, 0, 0)_{ENU}$. For example, the initial orientation stipulates that $+x_{LLC}$ is aligned east and $+y_{LLC}$ is aligned north.

An intrinsic rotation of the $(x, y, z)_{LLC}$ coordinates about the global origin, from the specified initial orientation, is carried out in the order of yaw-pitch-roll (i.e., a *Z-Y-X* rotation),

$$R = R_{X_g} \cdot R_{Y_g} \cdot R_{Z_g}, \tag{5}$$

such that,

$$\begin{bmatrix} x_{ENU} \\ y_{ENU} \\ z_{ENU} \end{bmatrix} = R \cdot \begin{bmatrix} x_{LLC} \\ y_{LLC} \\ z_{LLC} \end{bmatrix}. \tag{6}$$

The global easting, northing, and elevation coordinates, $(E, N, U)_{LLC}$, are computed via translation of the rotated point cloud, $(x, y, z)_{ENU}$, from the global coordinate origin, $(0, 0, 0)_{ENU}$, to the coordinates of the origin of the puck, following

$$\begin{bmatrix} E_{LLC} \\ N_{LLC} \\ U_{LLC} \end{bmatrix} = \begin{bmatrix} x_{ENU} \\ y_{ENU} \\ z_{ENU} \end{bmatrix} + \begin{bmatrix} E_{puck} \\ N_{puck} \\ U_{puck} \end{bmatrix}, \tag{7}$$

where $(E, N, U)_{puck}$ are the surveyed easting, northing, and elevation coordinates of the puck origin, respectively, relative to the global coordinate system.

## 3. Results

### 3.1. Background and Study Site

The DUring Nearshore Event eXperiment (DUNEX) was a collaborative experiment involving multiple academic institutions and federal agencies in the Outer Banks of North Carolina to improve understanding and prediction of storm processes. In association with DUNEX 2020, a pilot field campaign was conducted between 16–18 November 2020 at the U.S. Army Corps of Engineers (USACE) Field Research Facility (FRF) in Duck, North Carolina. The pilot study main objectives were to quantify the ability of the prototype LLC LiDAR system in measuring (i) beach profiles and morphodynamics, (ii) free-surface elevation over time, and (iii) wave runup (cross-shore distance and elevation) in a real-world,

sandy environment with various wet/dry and irradiance/natural light conditions. Beach profiles, morphology, and wave runup data from the permanently installed FRF DUNE LiDAR scanner were used in the comparative analysis for items (i) and (iii) [15,24,25]. The angular resolution of the FRF DUNE LiDAR is 0.025°. The FRF uses a coordinate system unique to the facility, with $x_{FRF}$ in the cross-shore direction and positive offshore, $y_{FRF}$ is positive at 18.1465° west of true north (Figure 2), and $z_{FRF}$ is vertical (earth-normal) following the right-hand-rule and uses the North American Vertical Datum of 1988 (NAVD88). The tidal environment in the region is characterized as mesotidal and semi-diurnal.

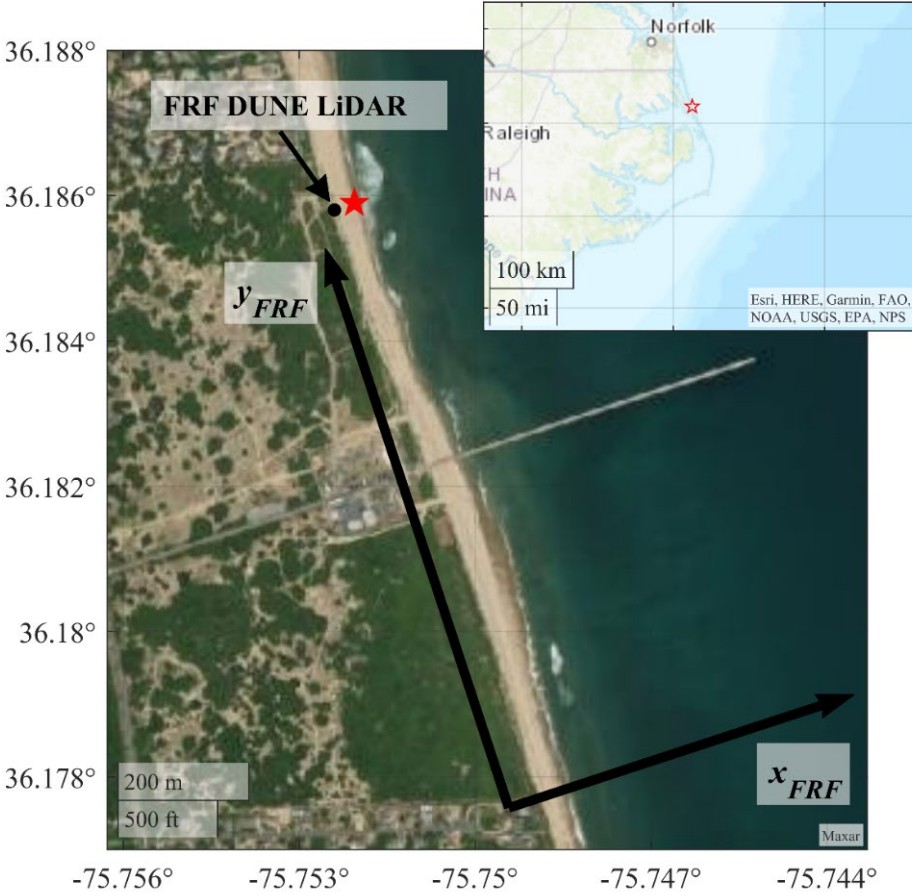

**Figure 2.** Field location along the Outer Banks of North Carolina, at the USACE FRF in Duck, NC. The local FRF coordinate system ($x_{FRF}$, $y_{FRF}$) is overlain on the satellite view. The location of the FRF DUNE LiDAR (black circle), and the approximate locations of the LLC LiDAR installations (red star) are shown. The $z_{FRF}$ coordinate is positive following the right-hand rule.

*3.2. Setup and Experiment Description*

Two series of tests were conducted between 16–18 November 2020. Series B encompassed the evening of 16 November to the morning of 17 November, and Series C spanned the evening of 17 November to the morning of 18 November. For each Series, an LLC LiDAR was mounted roughly 2 m above the beach (Figure 3), secured to the top of a galvanized steel pole using strut channels, vibration dampening routing clamps, and ratchet straps (Figure 1; right panel). Four guy wires were attached to the top of the pole and anchored into the beach during Series C to minimize pole vibrations upon wave impact during wave runup—vibrations that would adversely affect the LiDAR point cloud quality and accuracy. The installation sites during Series B and C were located roughly 3 m southeast (in the longshore direction) of the FRF DUNE LiDAR cross-shore transect, with the LLC LiDAR scanner installed on each pole such that the line-scan transect was near as possible to parallel with the FRF DUNE LiDAR transect (Figure 4). The angle between the

FRF DUNE LiDAR and LLC LiDAR transects was $1.01°$ ($0.81°$) for Series B (Series C). The cross-shore location of Site B (Site C) was $x_{FRF}$ = 72.30 m ($x_{FRF}$ = 83.93 m).

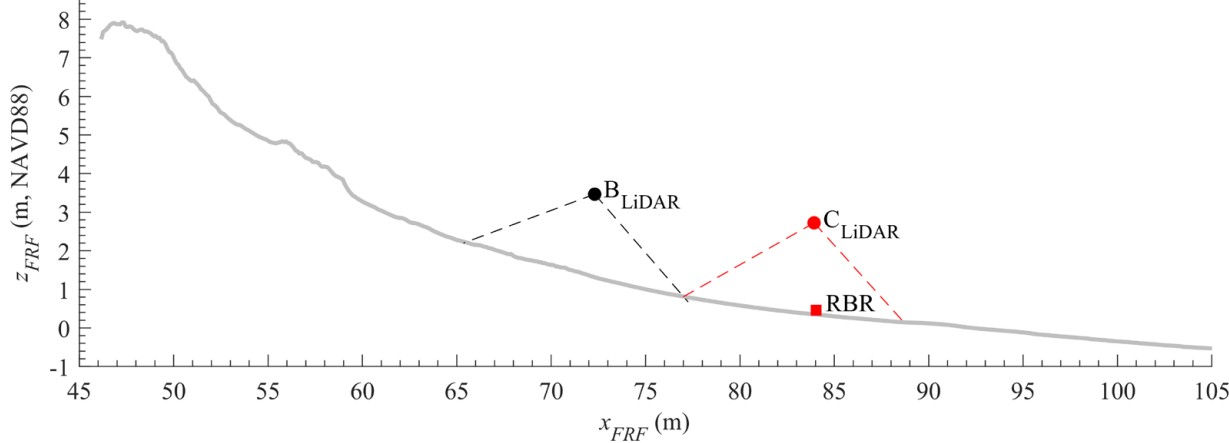

**Figure 3.** Profile view of the instrument installation locations, in FRF coordinates: LLC LiDAR (solid circles) and RBR$solo^3$ D|wave16 logger (square). The field of view of each LLC LiDAR is represented with dashed lines. Black (red) denotes Series B (Series C). The time-averaged beach profile measured by the FRF DUNE LiDAR over the deployment duration (16–18 November 2020) is shown for reference (thick gray).

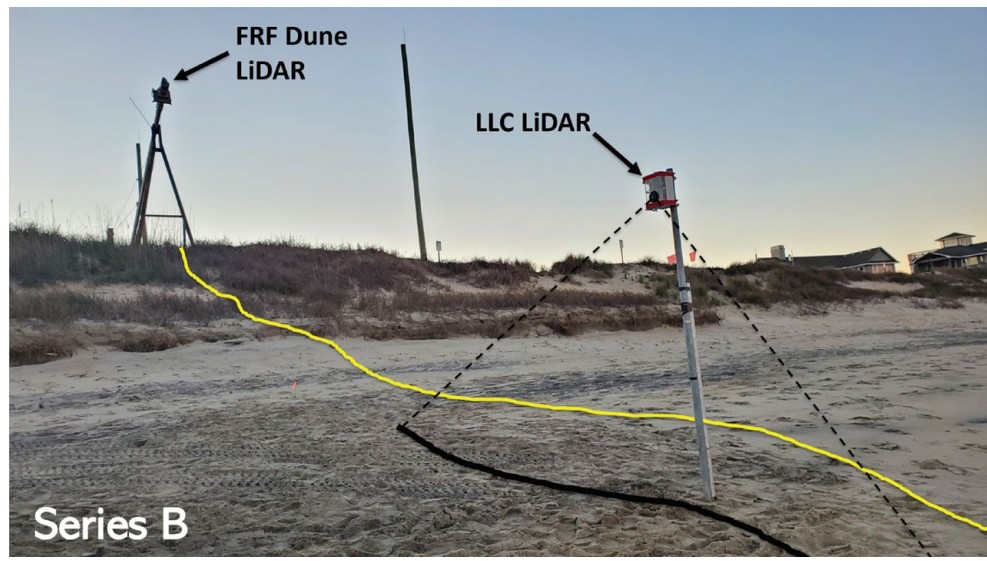

**Figure 4.** The FRF DUNE LiDAR and LLC LiDAR (Series B) deployed at the USACE FRF. The approximate LLC LiDAR profile range and field of view are drawn as solid and dashed black lines, respectively ($y_{FRF} ≈ 942$ m). The yellow line denotes the approximate cross-shore transect measured by the FRF DUNE LiDAR ($y_{FRF} ≈ 945$ m).

During each series, the LLC LiDAR scanned continuously for the first 25 min of every half hour at ~7 revolutions per second, with an average angular resolution, $d\theta$, of $1.3°$. A total of 35 (28) 25-min segments were collected during Series B (Series C). The FRF DUNE LiDAR performed line scans at 7.1 Hz during the first 30 min of every hour, yielding hourly time-averaged beach profiles and 30-min time series of wave runup excursion and elevation [15]. During Series C, waves and water levels were measured directly beneath the LLC LiDAR using an RBR$solo^3$ D|wave16 pressure logger sampling continuously at 16 Hz (hereafter, referred to as RBR). The RBR was installed at the base of the pole at Site C, ~0.15 m above the beach with the pressure diaphragm at 0.46 m, NAVD88 (Figure 3). The RBR

was deployed with the pressure diaphragm oriented downward following recommended installation protocol. The pressure diaphragm was not covered with a mesh filter and remained above the elevation of the bed during the entirety of the Series C deployment (i.e., it was never buried in sand).

Significant wave height, $H_s$ (peak wave period, $T_p$), measured by the permanently installed 8-m array at the FRF, ranged from 0.5 to 1.6 m (4 to 11 s) (Figure 5a,b). A medium swell event ($H_s = 1.6$ m, $T_p = 7$ s) arrived overnight during Series C, bringing an extended period of elevated water levels higher than forecast tide levels (Figure 5c). The study encompassed three complete high and two complete low tide cycles (high tides numbered 1 to 3 in Figure 5c). The tide range during the experiment was approximately 1.2 m. High tide #2 (#3) reached the cross-shore extent of Series B (Series C). The cross-shore extent scanned during Series C was in the swash regime during the rising and falling phases of high tide #3 but was in the inner-surf regime around the peak of high tide #3. The average beach slope in the Series B (Series C) cross-shore span was approximately 1:10 (1:30).

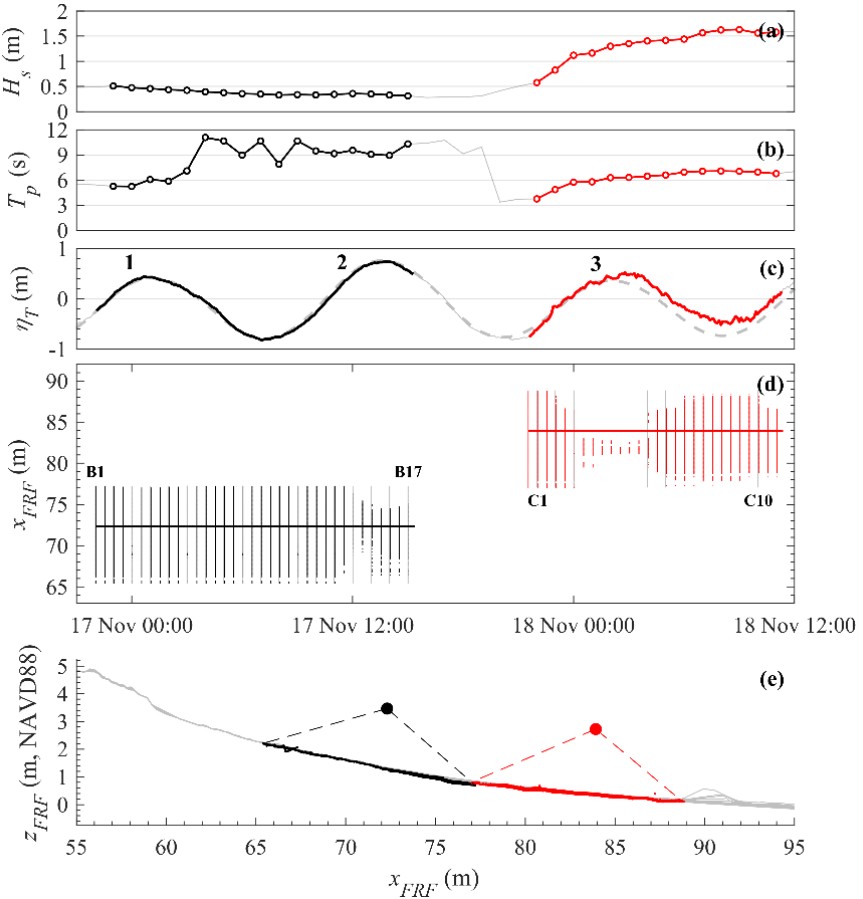

**Figure 5.** (**a**) Significant wave height, (**b**) peak wave period, and (**c**) measured (solid) and predicted (dashed) tidal water levels (NAVD88) at the USACE FRF in Duck, NC (NOAA Station ID: 8651370). The black (red) overlays indicate measured tidal water levels during the time span for the Series B (Series C) deployment. (**d**) Plan view time stack and (**e**) profile view of the half-hourly, time-averaged cross-shore beach profiles measured by the LLC LiDAR and the corresponding hourly FRF DUNE LiDAR profiles (gray). The FRF DUNE LiDAR cross-shore profile extents are truncated in (**d**) to demonstrate the spans over which statistics were computed in Table 1. Transects numbered in (**d**) correspond to Profile numbers in Table 1. The thick horizontal lines intersecting the profiles in (**d**), and the solid circles in (**e**), mark the locations of the LLC LiDAR puck. Dashed lines in (**e**) represent the field of view of each LLC LiDAR. Black (red) colors correspond to Series B (Series C) in all panels. Times in UTC.

The approximate heading of the earth-level $+x_{LLC}$ axis relative to true north was measured with a tilt-compensating Leica GS18-T RTK GPS antenna held over the LLC LiDAR, and the inclination angles were measured with a digital level placed on the top and sides of the LLC LiDAR enclosure during initial installation. The yaw angle was determined using

$$\alpha = 90 - \varphi, \tag{8}$$

where $\varphi$ is the compass heading of the earth-level $+x_{LLC}$ axis relative to true north. The easting, northing, and elevation coordinates of the puck origin, $(E, N, U)_{puck}$, were measured using the RTK GPS antenna (with tilt compensation enabled) on a 0.39 m-long survey pole, with the tip placed at the outer center of the puck. There is a 0.009 m offset in the $y_{LLC}$ direction from the outer surface of the puck (where the origin was surveyed) and the *actual* origin in the center of the puck. This minor offset was not accounted for. The measured yaw, pitch, and roll angles were $(\gamma, \beta, \alpha)_B = (18°, 1.5°, -2°)$ for Series B and $(\gamma, \beta, \alpha)_C = (18°, 1°, 0°)$ for Series C. The surveyed puck origin coordinates were $(E_{puck}, N_{puck}, U_{puck})_B = (901,697.168 \text{ m}, 275,002.612 \text{ m}, 3.465 \text{ m})$ for Series B and $(E_{puck}, N_{puck}, U_{puck})_C = (901,708.027 \text{ m}, 275,006.759 \text{ m}, 2.723 \text{ m})$ for Series C. Easting and northing were measured in the North Carolina State Plane (NCSP) projected coordinate system (NAD83, 2011), and elevation was measured in orthometric height (NAVD88) using the GEOID18 model.

### 3.3. LLC LiDAR Configuration

The system time on the Raspberry Pi in the LLC LiDAR was set before the start of each series by manually connecting the Raspberry Pi to a 4g wireless hotspot and pulling time from a pool of web-based NTP servers. The LLC LiDAR power was then maintained after setting the clock, because any power loss or reboot of the Raspberry Pi would result in clock drift of the system clock, since the Raspberry Pi does not have an on-board real time clock. Additional corrections for time offsets between the LLC LiDAR and RBR were required and are described in Section 3.4.5.

A second Raspberry Pi 3B+ was configured as a field router, to allow for wireless (local) communication between a WiFi-enabled device (e.g., tablet, smart phone, laptop) and the LLC LiDAR. The field router was powered by an external USB battery pack. The field router was configured to use Media Access Card (MAC) address filtering to assign a static IP address to the LLC LiDAR Raspberry Pi, which was configured in */etc/wpa_supplicant/wpa_supplicant.conf* to automatically connect to the field router network. A smart phone was also connected to the field router network, and a mobile application for Android devices called *RaspController* was used to execute custom commands, saved within the application, to configure and control the LLC LiDAR. Although, it is worth pointing out that any SSH client would have worked. A few examples of standalone applications with an SSH client include Terminal (macOS), PuTTY (Windows), Termux (Android), xTerminal (iPhone/iPad), as well as any built-in terminal on UNIX/Linix platforms. A schedule was set in the LLC LiDAR cron scheduler (crontab) to enable the Raspberry Pi WiFi chip at the top of every hour, for 10 min, after which it was disabled for 50 min. This allowed for remote SSH connection during the 10-min window, via the field router and a phone or tablet, to check the system health and/or modify sample settings, if needed, while optimizing power usage when WiFi was not needed. Additional features such as Bluetooth and the HDMI port were disabled during operation, to further reduce unnecessary power consumption.

Data collection was performed autonomously via setting a schedule in the crontab task scheduler to execute the runfile shell script at the top and bottom of each hour. A scan duration of 1500 s (i.e., 25 min) was used in this study. Data were written to the microSD card that also ran the operating system, Raspbian OS, on the LLC LiDAR Raspberry Pi. Data were offloaded from the LLC LiDAR via remote file transfer using WinSCP on a laptop connected to the field router network.

### 3.4. LLC LiDAR Data Treatment

#### 3.4.1. Point Cloud Georectification

For each 25-min span, several initial quality control steps and coordinate transformations were performed. Points within the blanking distance ($r \leq 15$ m) or beyond the manufacturer reported maximum reliable range ($r \geq 10$ m) of the RPLiDAR scanner were discarded. Points with a quality score of 0 were also discarded. Each point cloud was converted from polar coordinates to cartesian coordinates following Equation (1). Points above $z_{LLC} = 0$ m for Series B ($z_{LLC} = -1.1$ m for Series C) were discarded. A lower elevation was used for Series C because of clusters of points in the point cloud present due to the guy wires used to stabilize the pole on which the LLC LiDAR was mounted.

The cartesian coordinates, $(x, y, z)_{LLC}$, of each point cloud were transformed into NCSP (easting and northing), and orthometric height (NAVD88) using Equations (6) and (7), along with the rotation angles and puck origin coordinates described in Section 3.2. Finally, the easting and northing coordinates, $(E, N)_{LLC}$, of each point cloud were transformed into FRF coordinates, $(x_{FRF}, y_{FRF})_{LLC}$, by performing a 2D counter-clockwise rotation of 20.025292169° about 901951.6805, 274093.1562 m NCSP (see Figure 2), with $(z_{FRF})_{LLC}$ of each point being equal to the orthometric height, $U_{LLC}$ (NAVD88) (Figure 6). This coordinate transformation approach from NCSP to FRF coordinates is consistent with O'Dea et al. [15]. Hereafter, the subscript *LLC* will be dropped from the rotated and georectified point cloud coordinates. Clarification will be made any time point clouds from multiple LiDAR systems are displayed together in the FRF coordinate system. A sample 25-min georectified point cloud time stack from Series C is shown in Figure 6a, along with a 2-min segment in plan, profile, and front view (Figure 6b–d). Eight waves in a 2-min span can be identified in Figure 6b,d in areas with low point density (white regions), with the exception of the cross-shore location directly beneath the LLC LiDAR (red line), defined hereafter as 'nadir.' The edges between the lower density areas and the higher density point cloud demarcate the boundary between the 'dry' beach and the waves, representing the extent of swash excursion over time. For example, the wave runup excursion of the first wave (at 00:10:45 UTC) reaches $x_{FRF} = 81.2$ m (Figure 6d) and an elevation of $z_{FRF} = 0.48$ m (Figure 6d).

#### 3.4.2. Beach Profiles

Time-averaged beach profiles were computed on a cross-shore vector with uniform bin width of 0.10 m, matching the cross-shore grid resolution of the FRF DUNE LiDAR beach profiles [15] The cross-shore grids for the LLC LiDAR beach profiles spanned 65.4 m $\leq x_{FRF} \leq$ 77.3 m (76.9 m $\leq x_{FRF} \leq$ 88.9 m) for Series B (Series C), each with $m = 120$ bins. Time-averaged beach profiles were computed for five 5-min segments (zero overlap) within the 25-min scans to quantify temporal changes in beach elevation on the order minutes. Noise, non-beach features (e.g., people), and waves were removed from the georectified point clouds using an iterative standard deviation filter.

For each $i$th cross-shore bin, where $i = 1 \ldots m$, the mean, $(\mu_z)_i$, and standard deviation, $(\sigma_z)_i$, of the point elevations in the point cloud were computed over the $j$th temporal bin, where $j = 1 \ldots 5$. For each bin, all points with an elevation outside the range of $(\mu_z)_i \pm 3(\sigma_z)_i$ were removed. The mean and standard deviation of elevation were then re-computed, and all points with an elevation outside the range of $(\mu_z)_i \pm 3(\sigma_z)_i$ were again removed. This was repeated iteratively until there were no points remaining with an elevation outside the range of $(\mu_z)_i \pm 3(\sigma_z)_i$, after which, the iterative process was repeated for two standard deviations from the mean value, $(\mu_z)_i \pm 2(\sigma_z)_i$. The percent of non-NaN points discarded by the iterative filtering process varies across scans, ranging from <1% to 8.99%, with a mean of 2.67% for Series C. It can be assumed that a lower percentage of points were filtered out from Series B because of the lack of waves. The time-averaged beach profile elevation for each bin was defined as the average elevation of the filtered

point cloud. Time-averaged elevations were replaced with a not-a-number (NaN) in bins where

$$\left(\hat{N}_{pts}\right)_{i,j} \leq 6 \times 10^{-6}, \tag{9}$$

where $\left(\hat{N}_{pts}\right)_{i,j}$ is the proportion of non-NaN points in the $i$th cross-shore and $j$th temporal bin, relative to the total number of non-NaN points (time and space) in the entire filtered 25-min scan, such that

$$\sum_{j=1}^{n}\sum_{i=1}^{m}\left(\hat{N}_{pts}\right)_{i,j} = 1.0. \tag{10}$$

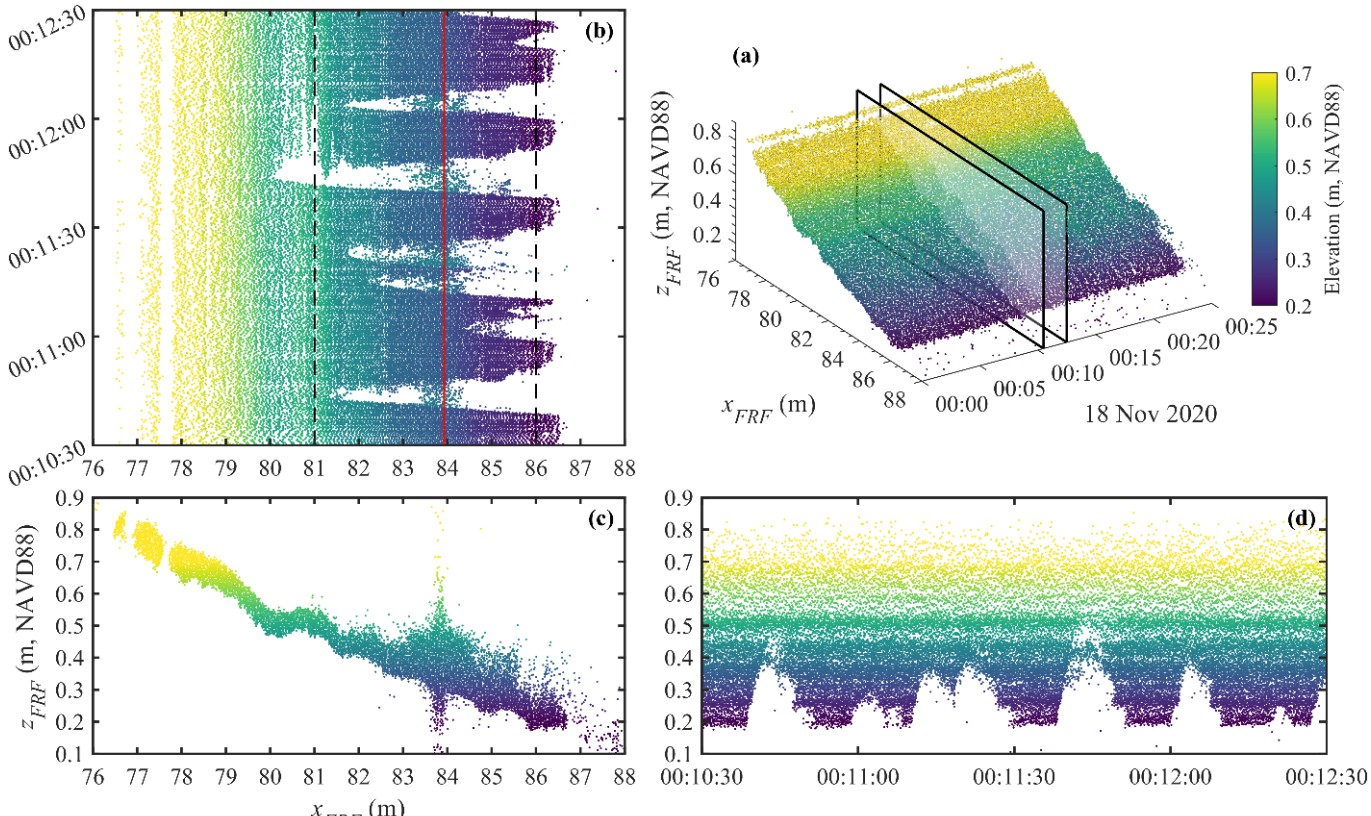

**Figure 6.** (**a**) Georectified point cloud time stack from a 25-min continuous LLC LiDAR scan during Series C (18 November 2020), with zoomed-in 2-min time stacks of scan data from three different perspectives: (**b**) plan view, (**c**) profile view, and (**d**) front view. The transparent surfaces in (**a**) mark the time bounds for the point clouds displayed in (**b**–**d**). The thick red line in (**b**) represents the cross-shore location of the LLC LiDAR puck (i.e., the nadir). The dashed black lines in (**b**) indicate the cross-shore range with suitable point density for the detection of wave runup with the LLC LiDAR. Times in UTC.

The condition in Equation (9) eliminates bins with low point density. Finally, any bins with $\sigma_z > 0.03$ m were also assigned a NaN value, regardless of point density, which generally occurred for bins with continuous wave action. Half-hourly beach profiles were computed for each 25-min scan as the time-average of the five 5-min profiles (Figure 7).

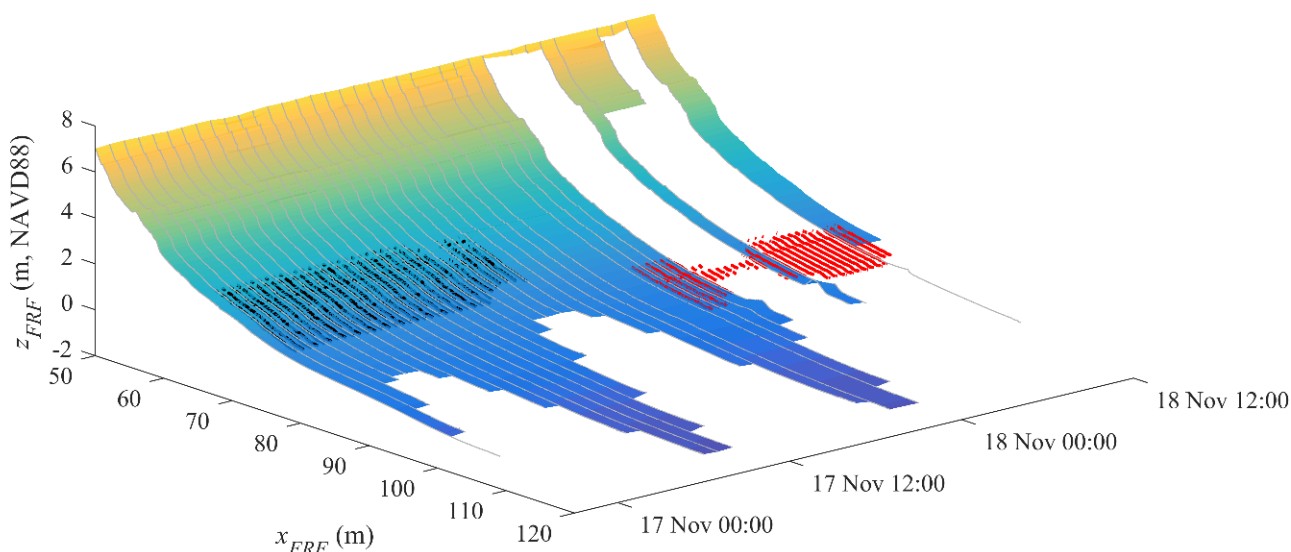

**Figure 7.** Time stack of half-hourly (hourly), time-averaged beach profiles measured by the LLC LiDAR (FRF DUNE LiDAR) for Series B (black) and Series C (red). The gray lines and colored surface are the FRF DUNE LiDAR hourly profiles. The thick black (red) lines running parallel to the time axis indicate the location of the LLC LiDAR puck for Series B (Series C).Times in UTC.

### 3.4.3. Gridded Surface

The cross-shore grids for wave runup detection spanned 67.3 m $\leq x_{FRF} \leq$ 77.2 m (78.9 $\leq x_{FRF} \leq$ 88.8 m) for Series B (Series C), each with $n = 33$ bins and 0.30 m bin width. Low point densities on the water surface away from the nadir of the LiDAR (Figure 6), and the need for higher temporal point density to interpolate onto a time vector of 8 Hz, necessitated the larger cross-shore bin width than was used in computing time-averaged beach profiles. For each $i$th cross-shore bin, where $i = 1 \ldots n$, the mean, $(\mu_z)_i$, and standard deviation, $(\sigma_z)_i$, of the elevations in the georectified point cloud were computed over the entire 25-min duration of each scan.

A filter was applied to the georectified point cloud of Series B with the stipulation that points with an elevation outside the range of $(\mu_z)_i \pm 3(\sigma_z)_i$ were removed (Figure 8a). The removal of a person walking through the scan plane of the LLC LiDAR is demonstrated in Figure 8a. A similar filter with different conditions was applied to the georectified point cloud of Series C such that points with an elevation above $(\mu_z)_i + 10(\sigma_z)_i$ or below $(\mu_z)_i - 3(\sigma_z)_i$ were removed (Figure 8b). A larger factor above the mean value ($10\sigma$ versus $3\sigma$) was used for Series C to reduce the removal of points associated with waves (i.e., the "free-surface")—points that are required for the estimation of wave runup (Section 3.4.4) and free-surface elevations (Section 3.4.5)—while still eliminating spurious and/or noisy points beneath the LLC LiDAR. The filter was applied once for both Series B and C (i.e., not iteratively).

The filtered point clouds were used to compute a gridded surface, $S$, for each series by applying a moving average (boxcar window width of 15 points, based on index, not time) to the point elevation in the $i$th cross-shore bin, then linearly interpolating onto a uniformly spaced time vector at 8 Hz and 25-min in duration. The moving average was computed assuming uniform temporal spacing between points in the filtered point clouds. The gridded surface, $S$, retained wave features and the beach elevation.

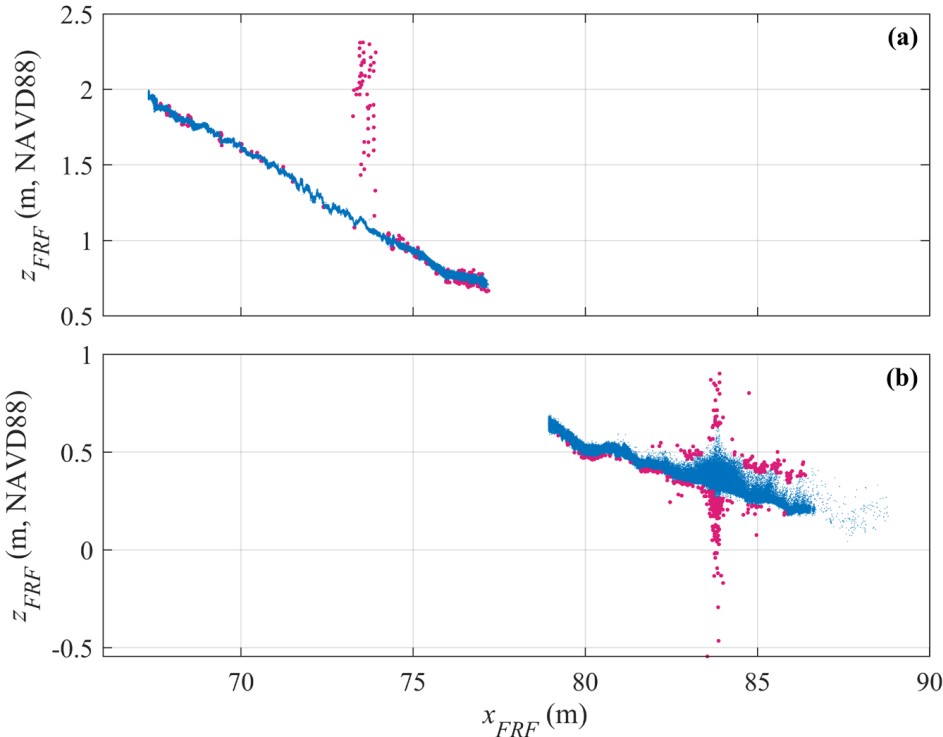

**Figure 8.** Profile view of georectified point clouds (magenta) versus filtered point clouds, *PC* (blue) for (**a**) Series B (scan start: 17 November 2020 00:00 UTC), and (**b**) Series C (scan start: 18 November 2020 00:00 UTC), each comprising a 25-min continuous scan.

### 3.4.4. Wave Runup

The gridded surface, $S$, was used in the estimation of the cross-shore excursion, $R_x(t)$, and vertical elevation, $R_z(t)$, of wave runup from LLC LiDAR data. Following O'Dea et al. [15] and Brodie et al. [24], fluctuations on the time scale of waves were isolated by subtracting a temporal moving minimum surface from $S$. The moving minimum surface was computed in time and at each cross-shore bin, with a boxcar window width, $T_{window}$. The moving minimum surfaces help identify time spans where little to no change occurs over time at a certain cross-shore location. A wave-isolated surface was created using,

$$\widetilde{S} = S - \mathbf{movmin}(S), \tag{11}$$

where **movmin** is the moving minimum function in MATLAB®. Spans where little to no change occurred (i.e., where waves were not present) are represented by near-zero values in $\widetilde{S}$. The leading and trailing edge of a wave signal will result in non-zero differences between the elevation surface and the moving minimum. A threshold value, $z_{thresh}$, was used to define the cross-shore extent of wave runup, $R_x$. The value of $z_{thresh}$ was a user-definable input based on where in the wave-isolated surface, $\widetilde{S}$, the presence of a wave front was detectable. For each time, the first exceedance of $z_{thresh}$ in $\widetilde{S}$, starting from the most onshore grid cell, is used to define $R_x(t)$.

Wave runup derived from LLC LiDAR data was compared to wave runup data obtained by the FRF DUNE LiDAR for two 25-min scan periods from Series C: during the (a) rising and (b) falling tide. Time synchronization was conducted by visually determining the time difference between LLC LiDAR data and FRF DUNE LiDAR data, and manually shifting the FRF wave runup time series by 26 s (20 s) for the rising (falling) tide scan, yielding the best temporal match with LLC LiDAR wave runup data throughout each 25-min scan period.

The $z_{thresh}$ and $T_{window}$ values were determined by minimizing the root-mean-square difference (RMSD) between $R_x(t)$ derived from the FRF DUNE LiDAR and LLC LiDAR,

while also maximizing the number of points in the LLC LiDAR-derived wave runup signal. Progressively lower RMSD values were achieved with increasing $T_{window}$ from 1 s to 14 s, above which, RMSD values stabilized. The optimal $z_{thresh}$ values were 0.015 m (0.016 m) for the rising (falling) tide phase. The optimal $T_{window}$ value was 14 s for both the rising and falling tide phases. The vertical extent of wave runup, $R_z(t)$, for both the rising and falling tide phases, was defined as the elevation of the intersection between $R_x(t)$ and gridded surface, $S$.

### 3.4.5. Free-Surface Elevation

RBR-derived free-surface elevations, $\eta_{RBR}$, were calculated from RBR pressure measurements (Pascals), $P_{RBR}$, using

$$\eta_{RBR} = \frac{(P_{RBR} - P_{atm})}{\rho g} + z_{RBR}, \tag{12}$$

where $P_{atm}$ is atmospheric pressure (Pascals), $\rho$ is the density of seawater (=1024 kg·m$^{-3}$), $g$ is gravitational acceleration (=9.81 m·s$^{-2}$), and $z_{RBR}$ is the surveyed elevation (m, NAVD88) of the RBR pressure diaphragm. Atmospheric pressure was measured by a permanent FRF weather station on site. Values of $\eta_{RBR}$ corresponding to $P_{RBR} - P_{atm} < -0.01$ dBar were assigned NaN. The spans where the RBR was measuring air pressure needed to be removed from the signal before making statistical comparisons with LLC LiDAR free-surface elevations. The moving minimum method described in Section 3.4.4 was applied to the RBR free-surface signal with $z_{thresh}$ = 0.01 m and $T_{window}$ = 6 s. This approach served to preserve as much of the leading and trailing edges of the wave signal as possible, while also removing the spans where only air was being measured.

LLC LiDAR surface elevations, $\eta_{LLC}$, were extracted from gridded surface, $S$ (Section 3.4.3), at the bin encompassing the cross-shore location of C$_{LiDAR}$ and the RBR (Figure 3). This cross-shore bin, centered at $x_{FRF}$ = 83.88 m, was directly beneath the LLC LiDAR ($x_{FRF}$ = 83.93 m) and may also be referred to as the nadir bin. A 0.3-m wide cross-shore bin, with 1.3° angular resolution and 2.4 m mount height, corresponds to a maximum of 5 points within the bin for each rotation. In some cases, zero or one points were measured within the bin during a rotation, usually during wave breaking directly within the nadir bin. Explicit distinction of 'free-surface' (i.e., the air-water interface) is not made for $\eta_{LLC}$ here, because the LLC LiDAR nadir bin was located in the swash zone for the majority of high tide #3 (Series C). Therefore, when waves receded seaward, beyond the field of view spanned by the nadir bin, the LLC LiDAR was sampling the beach surface elevation, with no free-surface present. The term 'free-surface' will only be used explicitly for $\eta_{LLC}$ when comparing RBR- and LLC LiDAR-based measurements, since the RBR only measured water levels.

The LLC LiDAR and RBR free-surface elevation signals required alignment in time to perform statistical analysis for validation. RBR-derived free-surface elevations were downsampled via linear interpolation onto a datetime vector with $dt = 0.125$ s (8 Hz)—the same time interval as $\eta_{LLC}$. A cross-correlation function was computed between the RBR and LLC LiDAR signals using **xcorr** in MATLAB®. The cross-correlation analysis was carried out using un-filtered $\eta_{RBR}$ directly computed from Equation (12), due to the requirement of zero non-NaN values in both signals when using the **xcorr** function. The phase lags corresponding to the top ten normalized cross-correlation scores were applied to the $\eta_{LLC}$ time vector. The ten time-shifted $\eta_{LLC}$ signals were plotted against $\eta_{RBR}$, one at a time, starting with the time shift corresponding to the highest normalized cross-correlation score. Visual inspection was used to determine the phase lag that corresponded to the most accurate alignment with the $\eta_{RBR}$ signal. When numerous phase lags were similar (e.g., only different by fractions of a second), the phase lag with the lowest corresponding root-mean-square error was used. This process was repeated for each 25-min time span of LLC LiDAR data with enough free-surface elevation data at the nadir to adequately perform a cross-correlation analysis. Nine LLC LiDAR time spans during high tide #3 (starting at 18 November 00:30 and ending at 05:25 UTC) had long enough $\eta_{LLC}$ signals, with enough

overlapping free-surface elevations with $\eta_{RBR}$, to reasonably establish a time shift. The time shift was applied to the datetime vector of each LLC LiDAR 25-min segment separately, if applicable. Time shifts between the LLC LiDAR and RBR free-surface elevations ranged from 7 to 12 s.

## 4. Discussion

### 4.1. Beach Profiles and Morphology

The ability of the LLC LiDAR to accurately measure beach profiles and morphology over time was validated by comparing time-averaged LLC LiDAR beach profiles with time-averaged beach profiles measured by the FRF DUNE LiDAR (an already-validated tool). Time-averaged beach profiles are computed from 30-min continuously sampled FRF DUNE LiDAR line-scans using the following approach. First, the cross-shore field of view is separated into two sections delineated based on maximum variance in reflectance: dry beach and swash/inner surf zones. Point cloud density is computed in 10 cm$^2$ bins, and points with elevations higher than 0.10 m above the elevation of maximum point density were removed. The FRF DUNE LiDAR point cloud in the swash/inner surf zone span is filtered using a series of empirically defined conditions tuned specifically for the FRF DUNE LiDAR location [15]. The mean elevation of the remaining points in the filtered point cloud represents the time-averaged beach profile elevation The LLC LiDAR is unable to delineate swash/inner surf zones from the dry beach based on reflectance; therefore, a single filtering approach was used for all cross-shore bins.

The 30-min time-averaged profiles measured by the FRF DUNE LiDAR were interpolated onto the same cross-shore vector as Series B and C for statistical analysis. Each 25-min time-averaged LLC LiDAR beach profile was compared against the FRF DUNE LiDAR beach profile with the nearest scan start time (Figure 5d). The first and last LLC LiDAR profile used for comparison during both Series B and C are labeled in Figure 5d, and these profiles correspond to the profiles as numbered in Table 1. During Series B, 17 LLC and FRF DUNE LiDAR beach profiles were used for statistical comparison. Data gaps in FRF DUNE LiDAR profiles resulted in only six overlapping LLC and FRF DUNE LiDAR profiles being available for comparison during Series C (C1, C2, C3, C6, C7, and C10). Time spans during Series C without FRF DUNE LiDAR data are visually represented as blank (white) spans landward of $x_{FRF} = 80$ m in the colored surface in Figure 7. The maximum cross-shore range capable of being measured with the LLC LiDAR in typical coastal field conditions is limited by (i) the height at which the scanner can be stably mounted and (ii) the oblique angle between the scanner and the beach surface, which is a function of the beach slope. The approximate height of the LLC LiDAR puck origin above the beach during Series B (Series C) was roughly 2.2 m (2.4 m), resulting in a reliable cross-shore range for measuring beach profile elevations approximately 12 m wide, spanning 65 m $< x_{FRF} <$ 77 m (77 m $< x_{FRF} <$ 89 m).

The RMSD values between LLC LiDAR and FRF DUNE LiDAR-derived beach profiles are given in Table 1. The mean ($\mu_{RMSD}$) and standard deviation ($\sigma_{RMSD}$) of the RMSD values across all scans for Series B (Series C) were 0.045 m and 0.004 m (0.031 m and 0.002 m), respectively. The small values of $\sigma_{RMSD}$ indicates that the $\mu_{RMSD}$ is representative of differences across profiles within each series, which means the accuracy of the LLC LiDAR in measuring beach profiles can be validated to within a few centimeters. The minor differences may be partially explained by the 3-m longshore separation between the transects scanned by the LLC LiDAR and the FRF DUNE LiDAR. Slight longshore variations in beach profiles were visually observed on site, particularly during Series B, which had the higher $\mu_{RMSD}$ and $\sigma_{RMSD}$ values. Errors in point cloud georectification due to survey error and/or rotation angle measurement may also contribute to non-zero RMSD values.

**Table 1.** Statistical comparisons between LLC LiDAR and FRF DUNE LiDAR time-averaged beach profile elevations. Profile numbers are defined in Figure 5.

| Profile | Series B RMSD (m) | Series C RMSD (m) |
|---|---|---|
| 1 | 0.039 | 0.027 |
| 2 | 0.043 | 0.029 |
| 3 | 0.045 | 0.031 |
| 4 | 0.041 | - |
| 5 | 0.043 | - |
| 6 | 0.040 | 0.032 |
| 7 | 0.041 | 0.031 |
| 8 | 0.043 | - |
| 9 | 0.043 | - |
| 10 | 0.043 | 0.033 |
| 11 | 0.046 | |
| 12 | 0.046 | |
| 13 | 0.045 | |
| 14 | 0.055 | |
| 15 | 0.046 | |
| 16 | 0.043 | |
| 17 | 0.053 | |
| $\mu_{RMSD}$ | **0.045** | **0.031** |
| $\sigma_{RMSD}$ | **0.004** | **0.002** |

Figure 9 shows progressive (Figure 9a) and cumulative (Figure 9b) beach profile elevation changes over time, alongside the measured tide elevation (Figure 9c). Shades of red (blue) represent net accretion (erosion) for progressive bed level change. Five-minute temporal gaps between the 25-min surfaces indicate the duration when LLC LiDAR scans were not sampled. Series B encompassed high tides 1 and 2, while Series C encompassed high tide 3. The elevation of high tide #2 was greater than the elevations of both high tides #1 and #3. Progressive bed level changes, $\Delta z(x, t)$, were computed by differencing successive 5-min time-averaged beach profiles to identify where erosion or accretion occurred on short (5-min) time scales (Figure 9a). Progressive bed level changes were computed by differencing successive 5-min time-averaged profiles within a 25-min scan, without carrying over the final time-averaged profile from the previous 25-min scan, such that $\Delta z(x, t)$, over the first five minutes was zero. Cumulative bed level changes, $\Delta z_{cm}(x, t)$, for each series were computed by differencing each 5-min time-averaged beach profile from the first 5-min time-averaged beach profile of the entire series (Figure 9b).

Small progressive bed level changes, $|\Delta z| < 0.01$ m, were resolved with the LLC LiDAR, allowing for identification of cross-shore and temporal erosion and accretion. During all three high tides, numerous net onshore erosion coupled with net offshore accretion events occurred in the swash zone. For example, net erosion occurred between 74 m $< x_{FRF} < 75$ m in the scan beginning on 17 November at 00:00 UTC, while net accretion occurred between 75 m $< x_{FRF} < 77$ m, indicating net offshore sediment flux. Erosion occurred at between 82.5 m $< x_{FRF} < 86.5$ m while accretion occurred between 79 m $< x_{FRF} < 82.5$ m in the last scan period of Series C, indicating net onshore sediment flux.

One major implication of being able to measure even small-scale cross-shore bed level changes over time in the swash zone is that—with the assumption of zero gradient in longshore sediment transport rate—the 1D Exner equation,

$$\frac{\partial z}{\partial t} = -\frac{1}{\phi_b} \frac{\partial q_s}{\partial x}, \tag{13}$$

provides a mechanism to estimate the net cross-shore sediment transport rate, $q_s(x)$, over a period of time during which considerable bed level changes occurred. In Equation (13), $\phi_b$ is the sediment volume fraction (i.e., random packing volume fraction), typically taken as

$\phi_b \sim 0.60$ to $0.64$ for natural quartz beach sands. Consequently, LLC LiDAR beach profile and bed level change measurements provide the capability of quantifying cross-shore sediment transport processes in the swash zone. Future investigations whereby an LLC LiDAR is co-located with measurements of flow velocity and sediment concentration profiles will further validate the capabilities of the LLC LiDAR in net cross-shore sediment transport rate quantification.

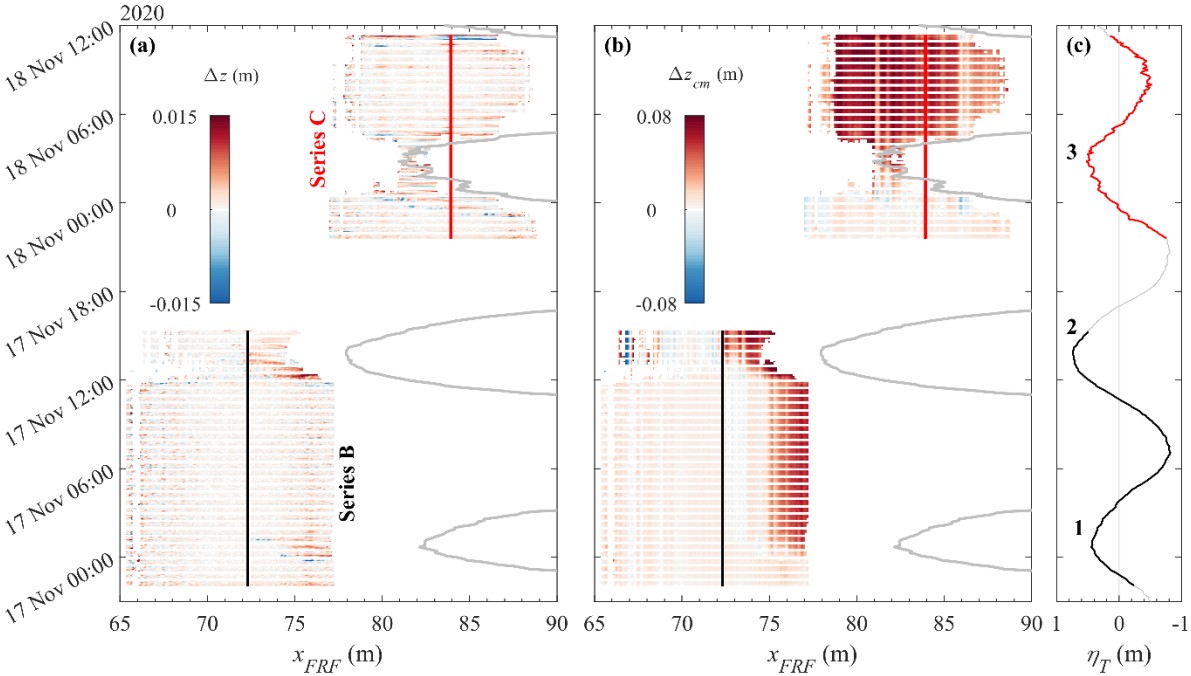

**Figure 9.** Beach profile elevation changes (**a**) progressively between successive profiles, $\Delta z$, and (**b**) cumulatively from the start of the series, $\Delta z_{cm}$; (**c**) measured tide elevation (NAVD88) – note the reversed horizontal axis. The thick black (red) lines running parallel to the time axes indicate the location of the LLC LiDAR puck for Series B (Series C). The thick grey line in (**a,b**) indicates where the measured tide elevation intersected the time-averaged FRF DUNE LiDAR beach profile. High tides in (**c**) are labeled 1, 2, and 3. Times in UTC.

*4.2. Wave Runup*

The ability of the LLC LiDAR to resolve elevation changes on the intra-wave time scale yields the opportunity to study small-scale morphodynamic processes, such as beach profile evolution (Section 4.1) and the cross-shore excursion and elevation of wave runup, which is a driver of beach profile evolution. Continuous time series of wave runup excursions and the associated changes in beach elevation allow for the quantification hydromorphodynamic processes, and during extreme events, the storm impact regime [10,30,31].

Figures 10 and 11 show the extent of wave runup in the cross-shore, $R_x(t)$, and elevation, $R_z(t)$, respectively, for two distinct tidal phases: rising tide (18 November 2020, 00:00 to 00:25 UTC, panels (a) and (b) of Figures 10 and 11) and falling tide (18 November 2020, 05:00 to 05:25 UTC, panels (c) and (d) of Figures 10 and 11). Statistics were only computed using data where both LLC- and FRF-derived runup values were within the optimal cross-shore zone for runup detection (81 m $\leq x_{FRF} \leq$ 86 m). The LLC LiDAR point cloud density was too low to reliably estimate the free-surface elevation outside of the optimal cross-shore zone for runup detection. LLC LiDAR wave runup extent, $R_x(t)$, and elevation, $R_z(t)$, compared reasonably well with the same quantities measured by the FRF DUNE LiDAR. RMSD values of wave runup estimates, $R_x$ and $R_z$, between the LLC LiDAR and FRF DUNE LiDAR were 0.542 m (76% of absolute differences, $|(R_x)_{LLC} - (R_x)_{FRF}|$, were below this value) and 0.039 m (72% of absolute differences, $|(R_z)_{LLC} - (R_z)_{FRF}|$, were below this value) for the rising tide, respectively, and 0.366 m (79% of absolute differences

were below this value) and 0.032 m (78% of absolute differences were below this value) for the falling tide, respectively. A large fraction of both rising and falling tide runup time series exhibited lower absolute differences (72% to 79%) than the RMSD, suggesting that short spans with large RMSD values—likely at the swash tongues—contributed largely to increasing the RMSD. The optimal cross-shore zone for runup detection is smaller than the reliable cross-shore range for measuring beach profiles because water is a less reflective and more scattering surface than sand, resulting in fewer returns per meter of cross-shore width (Figure 6b). In addition, linearly interpolating data with low temporal point density onto a time vector of 8 Hz could smooth out wave features present in the data, further reducing the detected elevation of already shallow swash tongues.

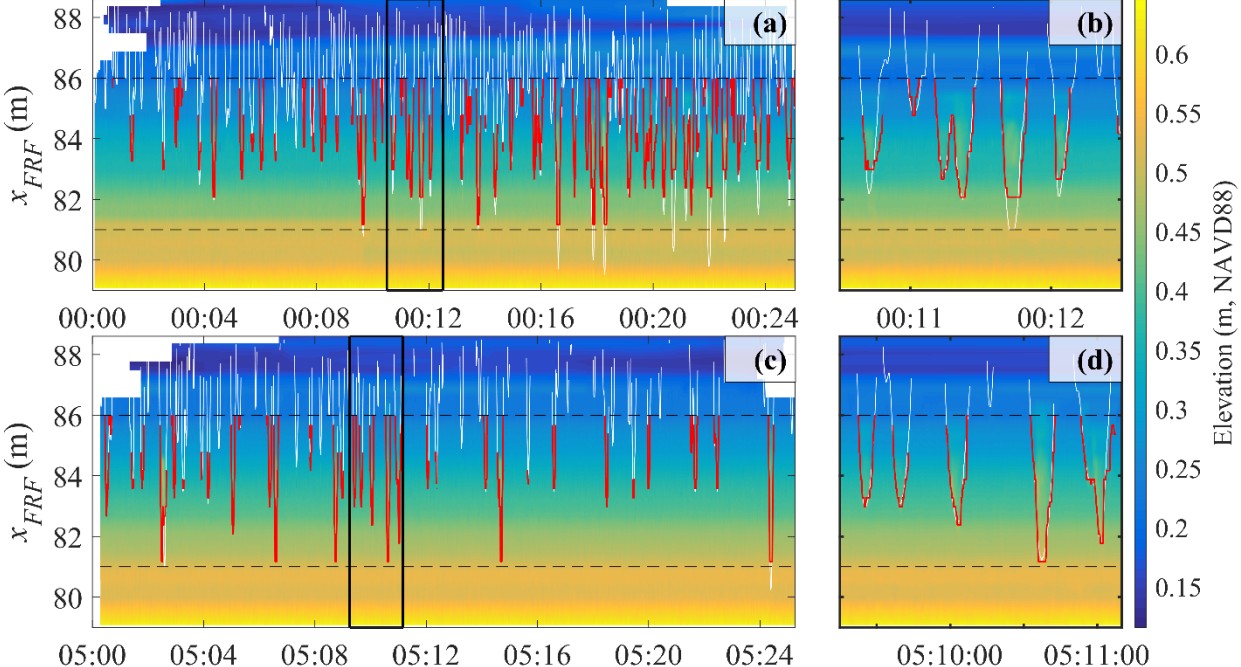

**Figure 10.** Cross-shore wave runup excursion, $R_x$, measured by the LLC LiDAR (red) and the FRF DUNE LiDAR (white) for two 25-min spans during Series C (18 November 2020): (**a**,**b**) rising tide, and (**c**,**d**) falling tide. The dashed black lines mark bounds of the optimal cross-shore zone for runup detection, beyond which, LLC LiDAR wave runup estimates were unreliable. The colored surface is the LLC LiDAR-derived surface elevation, $S$. The thick black boxes in (**a**,**c**) define the time spans expanded in (**b**,**d**). Times in UTC.

Some swash tongues within the optimal cross-shore zone for runup detection of the LLC LiDAR (e.g., at 00:11:45 in Figure 10 panels (a) and (b)) were not detectable likely due to the depth of the swash tongue not exceeding the empirical $z_{thresh}$ value in the runup detection algorithm, or decreasing point density on the water surface away from the nadir of the LiDAR (see Figure 6). The elevation reached by the maximum onshore extent of wave runup (wave by wave) that is exceeded 2% of the time is represented by $R_{2\%}$ [10,15]. Undetected swash tongues during peak wave runup and the limited range spanned by the optimal cross-shore zone for runup detection make a using a single prototype version of the LLC LiDAR poorly suited for determining $R_{2\%}$ over long durations.

Alternative deployment configurations that may improve the measurement and detection of wave runup may include deploying a cross-shore transect of multiple LLC LiDAR systems with overlapping fields of view, so wave fronts exceeding the most landward edge of the optimal cross-shore zone for runup detection from one scanner are detected by the subsequent scanner on the transect. Higher installation height, when feasible, would likely slightly improve the cross-shore extent of observable wave runup, since the LiDAR beam angle will be less oblique for a wider cross-shore range. The longer detection range

afforded by a cross-shore transect of multiple LiDAR units will make the LLC LiDAR better suited to resolve $R_{2\%}$, particularly during extreme events.

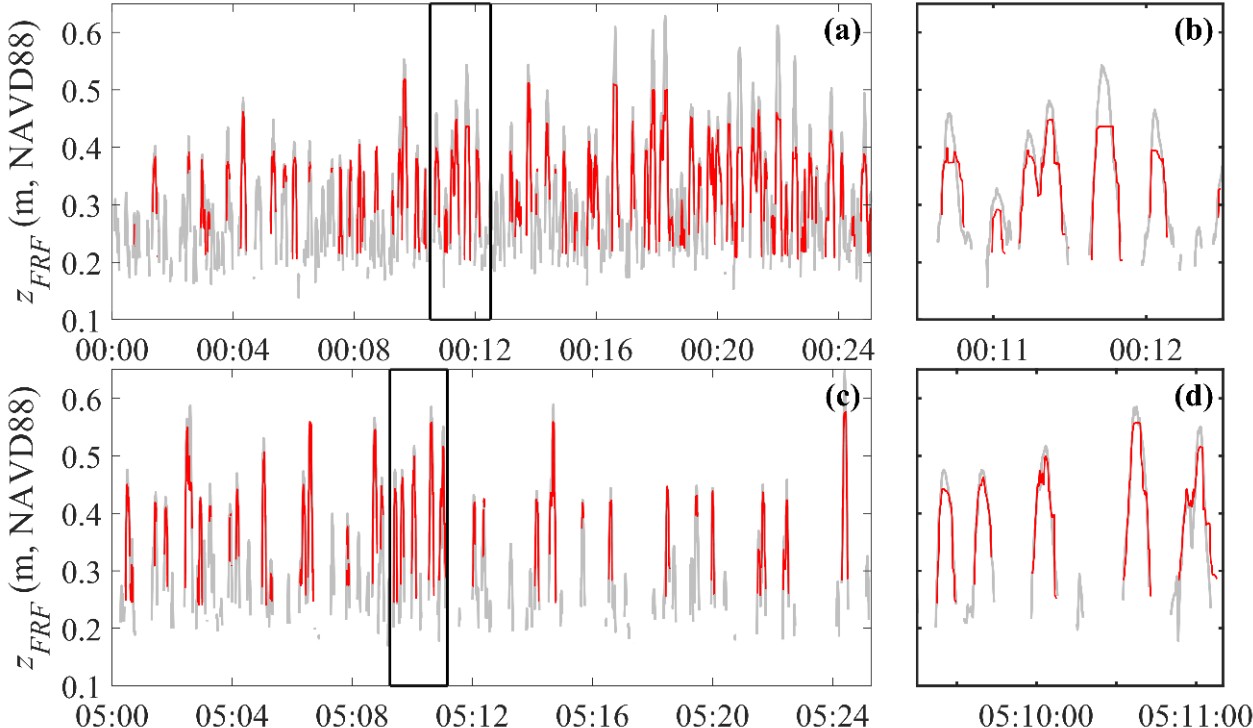

**Figure 11.** Elevation of wave runup, $R_z$, measured by the LLC LiDAR (red) and the FRF DUNE LiDAR (white) for two 25-min spans during Series C (18 November 2020): (**a,b**) rising tide, and (**c,d**) falling tide. The thick black boxes in (**a,c**) define the time spans expanded in (**b,d**). Times in UTC.

The current LLC LiDAR version uses the Slamtec RPLIDAR R5 model, which has an angular resolution of 1.3°. However, a newer Slamtec RPLIDAR R6 model uses upgraded hardware and software to achieve a higher angular resolution of 0.25°. Integrating the R6 model into the LLC LiDAR system will improve the cross-shore point cloud density, particularly during wave runup, which will be better suited for tracking wave fronts during uprush and backwash in the swash zone.

### 4.3. Free-Surface Elevation

The RMSD values between $\eta_{LLC}$ and $\eta_{RBR}$ for nine 25-min segments are shown in Figure 12a. The raw and filtered $\eta_{RBR}$ signals, as well as the $\eta_{LLC}$ free-surface elevation are shown in Figure 12b. Three zoomed-in 5-min segments of the same free-surface elevation data at mid-rising, high, and mid-falling tide are shown in Figure 13. The highest observed free-surface elevation at the nadir occurred just before 18 November 02:40 UTC (Figure 13b), reaching ~0.90 to 0.95 m, shortly before the time of high tide. The average RMSD between $\eta_{LLC}$ and $\eta_{RBR}$ was 0.027 m across all nine segments. The relatively low and consistent RMSD between RBR- and LLC LiDAR-derived free-surface elevations validates the LLC LiDAR as a viable tool for measuring free-surface elevation time series in the swash and inner-surf zones on a natural beach. An average RMSD of 0.027 m is near the margin of vertical survey accuracy between the two systems.

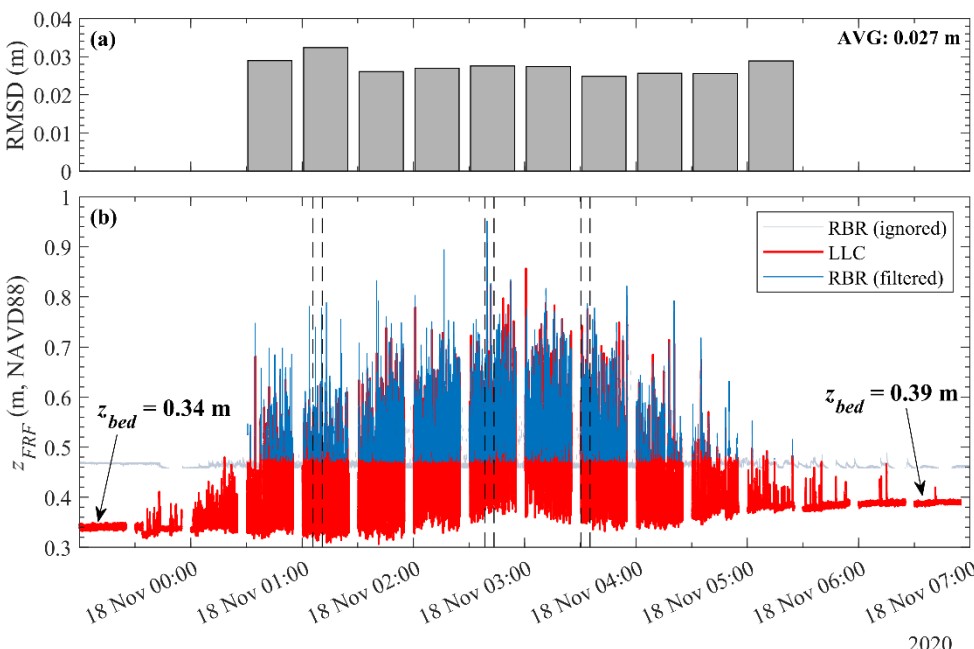

**Figure 12.** (**a**) Root-mean-square difference (RMSD) between RBR- and LLC LiDAR-derived free-surface elevation for each 25-min span. (**b**) Free-surface elevation during Series C (18 November 2020), high tide #3 (at $x_{FRF}$ = 83.88 m), measured by the RBR (blue) and LLC LiDAR (red). The light-blue curve indicates RBR data that were ignored for RMSD computation, due to water levels dropping below the RBR pressure transducer. The dashed black lines mark the time spans shown in Figure 13. Times in UTC.

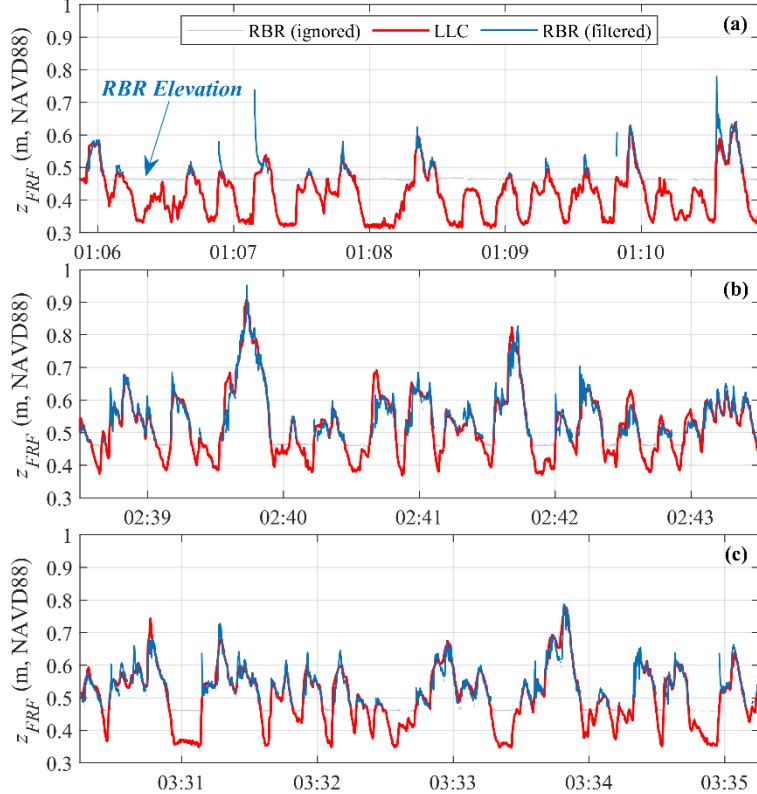

**Figure 13.** Three 5-min segments of beach and free-surface elevations derived from the LLC LiDAR and RBR (**a**) during mid-rising tide, (**b**) at high tide, and (**c**) during mid-falling tide. Times in UTC.

Equation (12) employed a hydrostatic assumption—a simplified approximation for depth using pressure measurements—which may not be proper for steep, bubbly, and/or foam-filled bore fronts typified by the swash zone [24]. When the small fraction of spans where $|\eta_{RBR} - \eta_{LLC}| > 0.10$ m were ignored (comprising 1% of overlapping LLC and RBR free-surface elevations), the average RMSD decreased to 0.024 m. An absolute difference exceeding 0.10 m occurred most often during spurious spikes in the pressure signal, or during swash bore arrival (e.g., Figure 13a), when dynamic pressure—likely the result of the downward orientation of the RBR and the lack of a geotextile-mesh cover over the pressure diaphragm—may cause a sharp spike in measured pressure that may not be representative of the actual free-surface elevation.

Bed level changes ($\Delta z_{bed}$) before and after high tide #3 were observed on the order of +0.05 m at the nadir (Figure 12b). This observation is corroborated by the cumulative bed level changes shown for Series C at the nadir (red line) in Figure 9b (17 November 23:30 to 18 November 07:00 UTC). RBR pressure loggers are unable to observe bed level changes, while the LLC LiDAR is effective in measuring both the free-surface elevation and bed level changes over time. Water levels cannot be measured by the RBR when the free-surface elevation drops below the elevation of the pressure diaphragm (see Figure 13a). Conversely, the LLC LiDAR system is capable of measuring near-uninterrupted time series of the free-surface, as well as small-scale (on the order of mm to cm) swash zone bed level changes (if any) between swash events, at both the nadir and along a cross-shore span of several meters.

A direct comparison of RBR- and LLC LiDAR-derived free-surface elevations ($n_{pts}$ = 40,738) is shown in a scatter plot, with the elevation of the RBR labeled for reference (Figure 14).

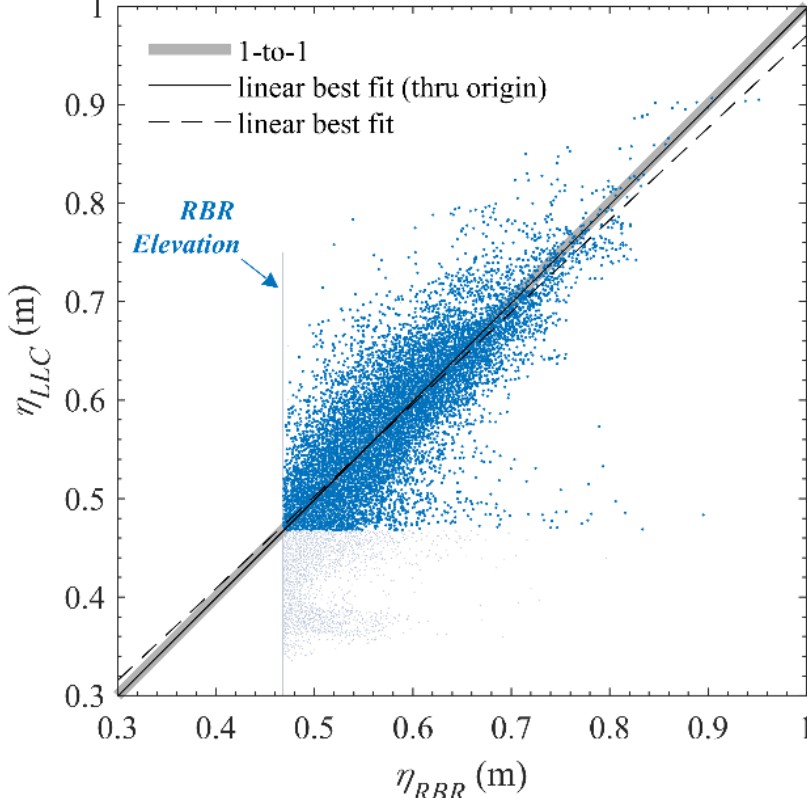

**Figure 14.** RBR- versus LLC LiDAR-derived free-surface elevations for nine 25-min segments during Series C (high tide #3). Dot colors follow Figure 12 legend. Linear best fits were performed via least-square-error regression.

The majority of the 1% of overall points with an absolute difference, $|\eta_{RBR} - \eta_{LLC}| > 0.10$ m, are visible as points below the 1-to-1 line ($0.6 \leq \eta_{RBR} \leq 0.9$), demonstrating that in general, the spikes and spurious points were either RBR over-estimation or LLC LiDAR under-estimation of the free-surface. The linear best fit least-square-error regression curve through LLC LiDAR- and filtered RBR-derived free-surface elevations (dashed line; Figure 14) yielded

$$\eta_{LLC} = 0.935\eta_{RBR} + 0.035, \tag{14}$$

where $\eta_{LLC}$ and $\eta_{RBR}$ are in meters, with a coefficient of determination, $R^2$, of 0.80. When absolute differences greater than 0.10 m were ignored, the slope (*y*-intercept) increased (decreased) to 0.957 (0.023 m), yielding $R^2 = 0.85$, indicating an overall better fit. If RBR-derived free-surface estimates are assumed to be accurate, then the ideal slope of the best fit would be 1.0 and the *y*-intercept would be 0.0. A slope less than 1.0 and *y*-intercept greater than 0.0 for the best fit in Equation (14) indicates slight bias of higher LLC LiDAR free-surface estimates for lower water levels, and higher RBR free-surface elevations for higher water levels. Forcing the least-square-error regression through the origin, using all $\eta_{LLC}$ and $\eta_{RBR}$ values (dark blue dots; Figure 14), yielded a slope of near unity (solid line; Figure 14),

$$\eta_{LLC} = 0.9977\eta_{RBR}, \tag{15}$$

where $\eta_{LLC}$ and $\eta_{RBR}$ are in meters, with an $R^2$ of 0.80. Overall, despite the LLC LiDAR-derived free-surface elevations being measured over a span of 0.30 m in the cross-shore, while the RBR-derived free-surface elevations are point measurements, reasonable agreement between the two different methods was achieved.

*4.4. Future Design Recommendations*

This field study was carried out as a proof-of-concept demonstration to determine the viability of fully standalone, low-cost LiDAR scanning technology, using an integrated systems approach, in measuring beach profile morphology, water levels, and wave runup. The goal was not to design a field-ready, storm-capable, multi-domain suitable LiDAR scanning system. As such, numerous features that would be desired in a 'more polished' design were necessarily ignored during prototype development. However, provided that the pilot field campaign yielded promising results, numerous future recommendations in mechanical, hardware, and software design for a future, more robust and scalable LLC LiDAR v2 model are outlined in this section.

The prototype LLC LiDAR as depicted in Figure 1 is neither waterproof nor ruggedized. An opening is visible between the lid and the RPLiDAR puck. As such, the prototype system is not suitable for extended coastal field studies due to risk of water intrusion from rain and damage to electronic components from corrosion caused by sea spray and air humidity. Therefore, the prototype LLC LiDAR system described herein is limited to short-term field applications with no rainfall in the forecast, or to wave flume laboratory studies studying beach morphodynamics (e.g., [32]). A fully weatherproof and ruggedized enclosure will involve developing a custom lid with a glass panel for the LiDAR laser pulse to emit. A rugged, custom lid can be 3D printed with a range of methods such as stereolithography (SLA) or selective laser sintering (SLS), which provides flexibility in the design shape of the lid and glass section while reducing the need for multiple parts to be assembled for the lid, thereby reducing risk for water intrusion. Furthermore, the glass must be transparent to the 785 nm wavelength of the RPLiDAR scanner. Curved glass will likely be difficult to manufacture and more expensive in low quantities; therefore, an elongated, thin, flat piece of glass is more desirable in a low-cost version of an upgraded LLC LiDAR system. One major drawback of a fully enclosed system with a glass panel is that salt creep and water droplets will adhere to the glass over time, which will require frequent cleaning to maintain high quality LiDAR point cloud data (i.e., eliminate refraction of the laser pulse through water droplets). This drawback will limit the possibility of long-term, standalone deployments; however, the current prototype LLC LiDAR is more limited

in deployment duration and conditions than a fully enclosed system would be. Finally, refraction of the laser beam through the glass will require calibration to obtain accurate ranging from the RPLiDAR scanner.

To be a viable tool for measuring hydro-morphodynamics during extreme events, both the enclosure and the mounting system must be fully ruggedized. That is, the LLC LiDAR should be able to withstand potential wave strikes without risk of water intrusion, physical damage to components, or altering of orientation. Piers or other coastal infrastructure will serve as excellent installation options for future deployments under extreme conditions. To be a viable tool for measuring hydro-morphodynamics over extended durations (weeks to months), the enclosure must be able to sustain long-term exposure to low and high air temperatures ($-10$ to $40$ °C), and even higher temperatures within the enclosure (up to $50$ °C). The LLC LiDAR should not fail to operate in this range of conditions. Management and diffusion of internal heat generated by the Raspberry Pi CPU, battery, and rotating RPLiDAR motor will be critical to achieve operations in the range of conditions outlined here. However, the solution for heat management should not sacrifice the enclosure weatherproofing or ruggedness, nor the system battery life. For example, an air-circulating fan may be a poor solution because of the significant power draw from the battery required to operate the fan.

System hardware encompasses all necessary components to operate the system which do not fall under the mechanical category. Examples of hardware on the present LLC LiDAR system include the Raspberry Pi single-board computer, the RPLiDAR scanner, and the USB battery pack. Two critical components required for an upgraded system include a real-time clock (RTC) and an inertial measurement system (IMU). An RTC will improve the clock keeping capability, thereby reducing clock drift between external instruments such as wave gauges (e.g., Section 3.4.5) or current meters. The LLC LiDAR orientation was not measured during or after deployment for the field study described herein. An integrated IMU will provide more accurate information about the initial orientation of the LLC LiDAR system, as well as the orientation of the system throughout the duration of a deployment, serving to improve the georectification process. If there is any rotation and/or translation of the LiDAR during a storm, georectification of the raw point cloud is non-trivial, unless the rotation occurs directly about the origin of the IMU, which is unlikely. Therefore, a sturdy, durable deployment platform is strongly recommended for future deployments, in conjunction with inclusion of an IMU. New georectification and/or point cloud registration methods should be developed to account for circumstances where it will not be possible to directly measure the origin of the RPLiDAR puck when the system is fully enclosed.

The battery used in the prototype LLC LiDAR design did not have pass-through charging (i.e., the ability to provide power to a system while simultaneously being re-charged by an external power source), which prohibited solar panel integration for solar re-charging during the day. A pass-through charging capable USB battery pack should be incorporated into a future version (e.g., Voltaic V88 battery), which will extend the possible duration of a deployment for regions with high solar irradiance. Integration of a solar panel would allow for extended deployments in remote regions where daily access may be limited (e.g., undeveloped barrier islands).

LiDAR point cloud data was written to the microSD card on the prototype LLC LiDAR version. Writing data to a USB flash drive would make the system more robust and data transfer faster via removal of the flash drive to copy data to another computer or server. Data could be transferred remotely, in real-time if a cellular modem is included in an upgraded LLC LiDAR system. Real-time data transfer would reduce the risk of data loss during extreme events if the LLC LiDAR system were damaged or lost during impact. However, real-time data transfer would require increased power usage, which would require a larger battery and may yield a solar re-chargeable, real-time system impractical. In addition, raw LiDAR data file sizes are relatively large (200 MB/h), which would require an expensive cellular data plan. On-board data processing could reduce the file sizes for real-time data transfer.

## 5. Conclusions

A prototype Line-scanning, Low-Cost (LLC) LiDAR system was developed at UNCW and a pilot field application was conducted at the U.S. Army Corps of Engineers, Field Research Facility (FRF) in Duck, NC, USA. This field study was carried out as a proof-of-concept demonstration to determine the viability of fully standalone, low-cost LiDAR scanning technology, using an integrated systems approach, in measuring beach profile morphology, water levels, and wave runup. The goal of the prototype development was not to design a field-ready, storm-capable, multi-domain suitable LiDAR scanning system. Rather, the goal was to prove the technology could be useful in measuring valuable parameters that are elusive during storm events, such as beach morphology, water levels, and wave runup. The FRF afforded a real-world, sandy environment with various wet/dry conditions and irradiance/natural light conditions. The field campaign encompassed three full tidal cycles and two cross-shore spans from the backshore to the intertidal (swash and inner-surf zones). Beach profiles and wave runup measured by the LLC LiDAR were validated against the same quantities measured by the continuously sampling FRF DUNE LiDAR scanner at the USACE FRF.

Free-surface elevations measured by the LLC LiDAR (at the nadir) were validated against the free-surface elevations measured by an RBR*solo*$^3$ D | wave16 pressure logger. The relatively low, and consistent RMSD (0.027 m) between RBR- and LLC LiDAR-derived free-surface elevations, along with a high $R^2$ value of 0.80, validates the LLC LiDAR as a viable tool for measuring free-surface elevation time series in the swash and inner-surf zones on a natural beach.

The LLC LiDAR captured near-uninterrupted time series of beach profile and water level change on the order of mm to cm at time scales on the order of minutes to seconds, respectively, which could prove useful in quantifying swash-driven beach morphology, as well as the major changes driven by storm waves and surge during extreme events. Future studies are recommended, in conjunction with fluid velocity and sediment concentration profile measurements, to evaluate the capabilities of next-generation LLC LiDAR systems in quantifying net cross-shore sediment transport rate in the swash zone.

**Author Contributions:** Conceptualization, R.S.M. and C.S.O.; methodology, R.S.M. and C.S.O.; software, R.S.M. and C.S.O.; validation, R.S.M. and C.S.O.; formal analysis, C.S.O. and R.S.M.; investigation, C.S.O. and R.S.M.; resources, R.S.M.; data curation, R.S.M. and C.S.O.; writing—original draft preparation, C.S.O. and R.S.M.; writing—review and editing, C.S.O. and R.S.M.; visualization, C.S.O. and R.S.M.; supervision, R.S.M.; project administration, R.S.M.; funding acquisition, R.S.M. All authors have read and agreed to the published version of the manuscript.

**Funding:** This research received no external funding.

**Data Availability Statement:** Data from the FRF DUNE LiDAR system are publicly available on the CHL THREDDS data server (CHL TDS).

**Acknowledgments:** The authors would like to acknowledge the financial assistance of the U.S. Coastal Research Program DUNEX campaign and the USGS. Support by Pat Dickhudt and the USACE FRF support staff, QA/QC on FRF DUNE LiDAR wave runup data by Annika O'Dea and Kate Brodie greatly improved the scope of this manuscript. Shawn Mieras designed the original internal mounting plate and developed the original 3D CAD model renderings. Drew Davey contributed to initial field tests on local beaches in North Carolina. Two anonymous reviewers provided commentary that significantly improved the quality of the manuscript.

**Conflicts of Interest:** The authors declare no conflict of interest.

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
