# Peer review of "Beach Profile, Water Level, and Wave Runup Measurements Using a Standalone Line-Scanning, Low-Cost (LLC) LiDAR System"

_remotesensing, doi:10.3390/rs14194968_

Round 1

Reviewer 1 Report

This is a well-written and highly detailed paper, and was a pleasure to read. Lidar measurements of wave runup are usually expensive, and if cheap, not precise. This paper demonstrates a new system that could be both, and is a promising start to the future of near remote beach observations. I recommend the paper be accepted with minor revisions, most of which are just small clarifications that could make the paper more useful to similar future studies.

One of the larger points I think worth nuancing is: The introduction is highly focused on beach change brought about by storms and extreme conditions, and the necessity of obtaining measurements during those time periods. The framing of the introduction suggests that this system will eventually be used in these conditions. Can these sensors (or further improvements on these systems) weather the storms? Rain (not mentioned) is an issue (for this lidar, and all lidars). Wind and rapid beach change in the area where you’ve set up your sensor is an issue, both of which have little to do with the lidar system itself but with the installation. How to secure anything in a potential area of slump, etc. In the conclusion of the paper, Line 755 states “The goal was not to design a field-ready, storm-capable, multi-domain suitable LiDAR scanning system.” I disagree, and think this could be reframed, as this sentence is entirely contradictory to your introduction. What you’ve clearly laid out in the paper is a promising step forward to the development of a  “field ready, storm-capable, etc etc.” While it is not a complete system now, you have demonstrated its potential. If you instead plan to downplay the [future] storm capabilities of this system, you might then consider reworking the introduction to focus less on storm and storm impacts, but instead on the costs, size, setup time of other terrestrial lidar systems.

Smaller comments:

Line 69: You might also include LiDAR runup measurements, as it is in the discussion. 

Line 79: suggest changing “towards providing” to “to provide”

Line 90: What is the ISO rating of the lidar puck? Can it handle rain? (I suspect not, but it is still worth mentioning)

Line 117: “commands to and point cloud”...something is missing here

Line 133: 1.3 deg angular resolution is much much larger than other terrestrial systems - it is probably worth mentioning at some point the angular resolution of the dune lidar (which is somewhere near 0.02 I think, smaller than your standard deviation!!)

Line 164: What determines “poor” and “great” in the quality score? Is it the strength of the return? Are you able to interpret what this means from the data?

Line 223: FRF angle is 18 or 18.2 (or 18.x?) deg? I think the actual rotation angle might be more precise? The precision is 0.01 in Line 244 when discussing the difference between the dune and llc transects, so I’d suggest stating the exact angle if this level of precision is important.

Line 238: “minimize pole vibrations”: was this measured at all? How were the guy wires anchored in the sand? (And the pole as well, I suppose.)

Line 241: Why is dune captilized? 

Line 261: suggest adding that the RBR was oriented downward following the recommended installation protocol. I had to look this up to make sure it was normal! 

Line 273: I do not see elevated water levels higher than forecast in figure 5c, only the measured tide?

Line 279: The citation Chardon-Maldoando here doesn’t seem necessary unless you explain that you deployed it here for a reason described in the C-M paper.

Line 283: State NOAA station ID

Line 300(ish): Were the yaw, pitch and roll measured only at the ouset of the series? Is it safe to assume that they were the same at the end of the each series?

Line 320: use *a* Media Access Card

Line 325: suggest re-wording this sentence to “Any SSH client would have worked; a few examples…”

Line 341: When is the data offloaded? At the end of the series? Or during? (How much time does data offload take? If these were to be deployed in storms, having the remote connection would be pretty handy to ensure no data loss…but that might also slow collection?)

Line 348: Why were points above 0 and -1.1 discarded? 

Figure 6c: this is a pretty thick cloud in the vertical as shown - approx 10cm. What percent of the data is discarded during the interative filtering?

Line 380: This bin width is remarkably small, given the large angular resolution of the system.

Line 401: suggest remove “really” 

Figure 7: It is somewhat difficult to see where the thick black/red lines are in this 3D view. I’d suggest you delete them here, since you already have them shown in Fig 5d, e.

Line 438: Can you explain the 8 Hz time vector, when you are collecting things at ~7 Hz? (And the dune lidar also collects at ~7 Hz?)

Line 440: *The* gridded surface

Line 442: The gridded surface

Line 449: How well does the wave signal extraction work when there are many NaNs in the data? (I assume there are quite a few given the angular resolution.) 

Line 487: CLidar typo (?) Do you mean LLC lidar?

Line 520: It is interesting that there are such large time discrepancies between the instruments (here and with the dune lidar). Is it safe to assume that the time shift issues are coming from the LLC lidar? It appears that the cross-correlation is showing that the timing on the LLC lidar is somewhat variable (largely stemming from the linear interpolation from start to finish, and then putting everything on an 8 Hz time vector)? Or is this the RBR?

Line 562: Slight longshore variations where visually observed - 3m is actually quite a big separation distance and it is surprising to me that the mean bias is only 4.5cm. (The dune lidar does have alongshore scans as well, so if you really wanted to verify things you could extract the transect from the alongshore scans…)

Figure 3: State what “1,2,3” mean in the figure text. (It is said earlier, but worth noting again here.)

Line 596: the last scan period of C is so different from the other more gradually changing scans that it almost appears as though there is slight tilt of the steel pole (or device). How is the stability of the installation measured in the field? 

Section 4.2: How does the runup tongue compare if you assume all NaN intersections are water/sand boundary? (Figure 6a, labeled b, looks very similar to argus timestacks - I just wonder if it could be treated in a similar fashion except change intensity for 0 to 15 point quality scale or something similar)

Line 783: Github site is not accessible

Author Response

Reviewer 1,

Reviewer 2 Report

Remotesensing-1915206

Beach Profile, Water Level, and Wave Runup Measurements Using a Standalone Line-Scanning, Low-Cost (LLC) LiDAR System

Christopher S. O’Connor and Ryan S. Mieras

The work introduces a measuring system of bottom and water surface elevation for foreshore morphology, wave, and sediment processes. The authors test the system at FRF in Duck and discuss its ability and preciseness. The trial is interesting, and this report would benefit the coastal science community.

The reviewer is satisfied with the manuscript. It describes well the method, data, and results. Please consider the following in preparing the final manuscript.

1) Please explain the following of the proposed system:
a) Possible duration of the power supply
b) Limit of the data storage (I suppose the Raspberry Pi has enough space to store the data)
c) Installation of the system to record the processes under (extremely) high wave conditions

2) The swath width of the proposed system is not so large. Please show your idea to extend the coverage.

3) Line 755: Is the 'not' a typo?
The goal was not to design a field-ready, storm-capable, multi-domain suitable LiDAR scanning system.

Author Response

Reviewer 2,
